# DOUBLY ROBUST STRUCTURE IDENTIFICATION FROM TEMPORAL DATA

## ABSTRACT

Learning the causes of time-series data is a fundamental task in many applications, spanning from finance to earth sciences or bio-medical applications. Common approaches for this task are based on vector auto-regression, and they do not take into account unknown confounding between potential causes. However, in settings with many potential causes and noisy data, these approaches may be substantially biased. Furthermore, potential causes may be correlated in practical applications. Moreover, existing algorithms often do not work with cyclic data. To address these challenges, we propose a new doubly robust method for Structure Identification from Temporal Data (SITD). We provide theoretical guarantees, showing that our method asymptotically recovers the true underlying causal structure. Our analysis extends to cases where the potential causes have cycles and they may be confounded. We further perform extensive experiments to showcase the superior performance of our method.

## 1 INTRODUCTION

One of the primary objectives when working with time series data is to uncover the causal structures between different variables over time. Learning these causal relations and their interactions is of critical importance in many disciplines, e.g., healthcare (Anwar et al., 2014), climate studies (Stips et al., 2016; Runge et al., 2019a), epidemiology (Hernán et al., 2000; Robins et al., 2000), finance (Hiemstra & Jones, 1994), ecosystems (Sugihara et al., 2012), and many more. Interventional data is not often accessible in many of these applications. For instance, in healthcare scenarios, conducting trials on patients may raise ethical concerns, or in the realm of earth and climate studies, randomized controlled trials are not feasible.

In general, understanding the underlying causal graph using only observational data is a cumbersome task due to many reasons: i) observational data, as opposed to interventional data, capture correlation-type relations[1] instead of cause-effect ones. ii) unobserved confounders introduce biases and deceive the algorithms to falsely infer causal relations instead of the true structure, e.g., the existence of a hidden common confounder iii) the number of possible underlying structures grows super exponentially with the number of variables creating major statistical and computation barriers iv) the identifiability problem, since multiple causal models can result in the same observational distribution, thus making it impossible to uniquely determine the true structure. To overcome these problems and determine the underlying structure, additional assumptions are imposed. Typical assumptions include faithfulness, linearity of relations, or even noise-free settings, which limit the types of causal relationships that can be discovered (Pearl, 2000; Peters et al., 2017; Spirtes et al., 2000; Glymour et al., 2019). Almost all of these challenges [2] extend to the problem of identifying the underlying causal structure from observational time-series datasets.

Subsequently, in many instances, the emphasis is placed on particular target variables of interest and their causal features. Causal features of a target are defined as the set of variables that conditioned on them, the target variable is independent of the rest variables. Causal feature selection enables training models which are much simpler, more interpretable, and more robust (Aliferis et al., 2010a; Janzing et al., 2020). However, learning the causal structures between variables and a specific target is still a demanding task, and many current approaches for causal feature selection face limitations by making

---

[1]which capture how changes in one variable are associated with changes in another variable.

[2]The arrow of time assumption can slightly simplify a very limited number of cases. See Axiom (B) for definition.

unrealistic simplifying assumptions about the data-generating process or by lacking computational and/or statistical scalability (Yu et al., 2021; 2020). As an important example, under the faithfulness assumption, causal features are equivalent to the Markov Blanket (MB) of the target variable (Pearl, 2000). Consequently, numerous research efforts have been dedicated to recovering the MB of the target variable (Tsamardinos et al., 2003b;a; Pena, 2008; Aliferis et al., 2010a; Yaramakala & Margaritis, 2005; Aliferis et al., 2010b; Margaritis & Thrun, 1999; Aliferis et al., 2003; Borboudakis & Tsamardinos, 2019; Tsamardinos et al., 2019). Nonetheless, this assumption is confining and these algorithms generally face the curse of dimensionality statistically and/or computationally. These challenges become particularly amplified in the context of time series data, where the number of variables grows linearly with the length of the data trajectories. As a result, to mitigate these problems, additional assumptions, e.g., stationarity or no hidden confounders are included (Moraffah et al., 2021; Bussmann et al., 2021; Runge, 2018) and/or weaker notions of causality[3] such as Granger causality have been studied extensively (Granger, 1988; 1969; Marinazzo et al., 2011; Tank et al., 2018; Bussmann et al., 2021; Khanna & Tan, 2019; Runge, 2018; Hasan et al., 2023).

The existing causal feature selection from time-series data algorithms often assume some level of faithfulness or causal sufficiency (Runge et al., 2019b; Runge, 2018; 2020). Oftentimes, they overlook the presence of unknown confounding factors among potential causes (Moraffah et al., 2021). Moreover, most cannot adapt to cyclic settings (Entner & Hoyer, 2010), which is relatively ubiquitous in many domains (Bollen, 1989). Furthermore, many algorithms employ the popular vector auto-regression framework to model time-dependence structures (Bussmann et al., 2021; Lu et al., 2016; Chen et al., 2009; Weichwald et al., 2020; Hyvärinen et al., 2010), which again is restrictive. To overcome these problems and inspired by the debiased machine learning literature (Chernozhukov et al., 2018), we propose an efficient algorithm for doubly robust structure identification from temporal data.

**Our contribution.**

- We provide an efficient and easy-to-implement doubly robust structure identification from temporal data algorithm (SITD) with theoretical guarantees enjoying $\sqrt{n}$-consistency.

- We provide an extensive technical discussion relating Granger causality to Pearl's framework for time series and show under which assumptions our approach can be used for feature selection or full causal discovery.

- We provide theoretical insights demonstrating that our algorithm doesn't need faithfulness or causal sufficiency while allowing for general non-linear cyclic structures and also the presence of hidden confounders among the covariates for the task of finding the direct causes of the target variable. An important problem in many applications.

- In extensive experiments we illustrate that our approach is significantly more robust, significantly faster, and more performative than state-of-the-art baselines.

## 2 RELATED WORK

A longstanding line of work intends to tailor the existing causal structure learning and Markov blanket discovery for i.i.d. data to the temporal setting. To name a few, Entner & Hoyer (2010) adapted the Fast Causal Inference algorithm (Spirtes et al., 2000) to time-series data. While the approach shares the benefit of being able to deal with hidden confounders, it is not possible to account for cyclic structures. Runge et al. (2019b) introduced PCMCI, as an adjusted version of PC (Spirtes et al., 2000) with an additional false positive control phase which is able to recover time-lagged causal dependencies. PCMCI+ modified the approach further to additionally be able to find contemporaneous causal edges (Runge, 2020). LPCMCI extends the scope by catering to the case of hidden confounders (Gerhardus & Runge, 2020). Even though methods in this category are able to provide theoretical guarantees for learning the causal structure, SITD has several advantages over them: i) For all of these methods, the faithfulness assumption is a key ingredient while SITD does not need it. ii) These methods are based on conditional independence testing which is widely recognized to be a cumbersome statistical problem (Bergsma, 2004; Kim et al., 2022). Shah & Peters (2020) have established that no conditional independence (CI) test can effectively control the Type-I error for all CI settings. Moreover in practice, conducting many conditional independence tests from lengthy time-series is burdensome. iii) Even having access to a perfect conditional independence test

---

[3]weaker than Pearl's structural equation model (Pearl, 2000).

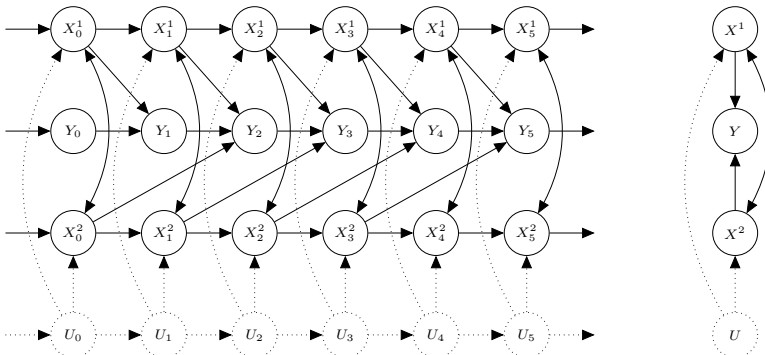

Figure 1: Example of a full-time graph (left), and the corresponding summary graph (right). Note that time series $X^1$ and $X^2$ causally influence the outcome $Y$ with different lags. The time series $U$ is unobserved, and it acts as a confounder for $X^1$ and $X^2$. Moreover, there is a cycle between the confoundings variables $X^1$ and $X^2$ of the outcome variable $Y$.

oracle, severe computational challenges exist (Chickering, 1996; Chickering et al., 2004)[4] (please refer to Appendix A.9 for a detailed comparison).

Another line of work relies heavily on the vector auto-regression framework. VarLiNGAM (Hyvärinen et al., 2010) generalizes LiNGAM (Shimizu et al., 2006) to time-series and similar to that it assumes a linear non-Gaussian acyclic model for the data. In the work of Huang et al. (2019), a time-varying linear causal model is assumed, allowing for causal discovery even in the presence of hidden confounders. More recently, deep neural networks are used to train vector auto-regression. Tank et al. (2018) adapted neural networks (named cMLP and cLSTM) for Granger causality by imposing group-sparsity regularizers. In a similar fashion, Khanna & Tan (2019) used recurrent neural networks. In another work, Bussmann et al. (2021) designed neural additive vector auto-regression (NAVAR), a neural network approach to model non-linearities. In contrast to previous works, they extract Granger-type causal relations by injecting the necessary sparsity directly into the architecture of the neural networks. This line of work is quite limited to ours as they consider confining structural assumptions over the underlying causal structural equations; details on these assumptions are discussed next to Axiom (A) in the following section.

The use of double robustness in causality problems has a long history mainly concentrated on estimating average treatment effect (Robins et al., 1994; Funk et al., 2011; Benkeser et al., 2017; Bang & Robins, 2005; Słoczyński & Wooldridge, 2018). Chernozhukov et al. (2018) introduced the DoubleML framework to achieve double robustness for structural parameters. Upon that, Soleymani et al. (2022) defined an orthogonalized score to infer the direct causes of partially linear models. Their approach is fast and allows for the assumption of complicated underlying structures but unfortunately, it is limited to only linear direct causal effects. While this was later extended to the non-linear case (Quinzan et al., 2023), we propose a doubly robust approach for identifying causal structures from temporal data under the general assumptions discussed below.

## 3   FRAMEWORK

### 3.1   PROBLEM DESCRIPTION

We are given i.i.d. observations from a univariate target time series $\boldsymbol{Y} := \{Y_t\}_{t \in \mathbb{Z}}$ and i.i.d. observations from a multivariate time series $\boldsymbol{X} := \{X_t^1, \ldots, X_t^m\}_{t \in \mathbb{Z}}$ of potential causes. We assume that the outcome time series is specified by some of the potential causes by a deterministic function with posterior additive noise, and no instantaneous effects. We can formalize this model as

Axiom (A)   $Y_T = f(\mathsf{pa}_T(\boldsymbol{Y}), T) + \varepsilon_T,$

with $\varepsilon_T$ exogenous independent noise and $\mathsf{pa}_T(\boldsymbol{Y}) \subseteq \{X_t^1, \ldots, X_t^m\}_{t \in \mathbb{Z}}$ is a subset of random variables of the multivariate time series $\boldsymbol{X}$. Note that the independence of the additive noise is important for identifiability. In fact, if there are dependencies between the noise and the history, then

---

[4]Learning Bayesian Networks with conditional independence tests is NP-Hard.

one might run into identifiability problems. We refer the reader to Appendix A.3 for a counterexample. We are interested in identifying the time series $\boldsymbol{X}^i$ that directly affects the outcome $Y$. That is, we wish to identify time series $\boldsymbol{X}^i$ such that it holds $X_t^i \subseteq \mathsf{pa}_T(\boldsymbol{Y})$ for some time steps $t, T$.

As discussed in Section 2, vector auto-regression methods enforce heavy structural assumptions on the underlying causal structural equations. NAVAR (Bussmann et al., 2021) assumes that each variable is influenced by its causal parents exclusively in an additive way and higher-order interactions among them are precluded. In mathematical terms, they follow

$$Y_T = \beta + \sum_{X \in \mathsf{pa}_T(\boldsymbol{Y})}^{N} f_X(X_{t-\kappa:t-1}) + \eta_t,$$

where $\beta$ is a bias term, $\kappa$ is the time lag, and $\eta_t$ is an independent noise variable. VarLiNGAM (Hyvärinen et al., 2010) takes a step further and uses much more restrictive assumptions on the functional form of the structural equations, confining to $f_X$ being a linear transformation of $X_{t-\kappa:t-1}$. These structural assumptions are quite limited as they do not include even simple functionals, e.g., $Y_t = X_{t-1}^1 \times X_{t-1}^2$ or $Y_t = \log(X_{t-1}^1 + X_{t-1}^2)$, whereas Axiom (A) is able to capture any interaction between the covariates and timestep $T$ which is a significant improvement over existing works.

**Additional assumptions.** In order to learn causes from observations, we further assume that

    Axiom (B)  there are no causal effects backward in time;

    Axiom (C)  there are no instantaneous causal effect between $\boldsymbol{Y}$ and any of the potential causes $\boldsymbol{X}^i$.

Both Axiom (B) and Axiom (C) appear in previous related work (see, e.g., Peters et al. (2013); Mastakouri et al. (2021); Löwe et al. (2022)). Note that these axioms allow for cycles and hidden common confounders between the potential causes. Axiom (B) is a natural assumption as a system is called causal when the output of the system only depends on the past, not the future (Peters et al., 2017; Pearl, 2000). Axiom (C) poses additional restrictions on the class of models that we consider, since instantaneous effects may be relevant in some cases and applications (Lippe et al., 2022a)[5]. However, without this assumption, it is impossible to learn causes *from observational data*. The necessity of Axiom (C) for causal discovery is a well-known fact (Peters et al., 2013)[6]. For the sake of completeness, we have furnished an example of this identifiability problem tailored to time-series data in Appendix A.4. For simplicity, we also consider the following assumption:

    Axiom (D)  the causal relationships are stationary, i.e., they are invariant under a joint shift of all the variables.

Axiom (D) again follows previous related work (Peters et al., 2013; Mastakouri et al., 2021; Löwe et al., 2022). This assumption is not strictly necessary for our analysis, but it is a general assumption that simplifies modeling. Our results can be extended to models that do not fulfill this axiom.

Our framework retains some degree of generality over previous related work. In fact, Axiom (A)-Axiom (D) allow for hidden confounding and cycles between the potential causes (see Figure 1), providing a more general framework than the full autoregressive model studied, e.g., by Hyvärinen et al. (2010); Peters et al. (2013); Löwe et al. (2022); Wu et al. (2020). Furthermore, in contrast to several previous works (Khemakhem et al., 2020; Gresele et al., 2021; Lachapelle et al., 2022; Lippe et al., 2022b; Yao et al., 2021), we do not assume that the variables $Y_T, X_T^1, \ldots, X_T^m$ are independent conditioned on the observed variables at previous time steps. Importantly, we also do not assume causal faithfulness in the full time graph, or weaker notions such as causal minimality.[7] This is a major improvement over some previous works, e.g., Mastakouri et al. (2021); Gong et al. (2022),

---

[5]An example of cases where time series exhibit instantaneous causal effects is given by dynamical systems (Mogensen et al., 2018; Rubenstein et al., 2016). In dynamical systems, a variable may instantaneously affect another variable of the model. Instantaneous effects have been studied in previous related work (Gong et al., 2022) but due to identifiability issues, they rely on stronger assumptions such as faithfulness.

[6]In general, Peters et al. (2013) show that causal discovery is impossible with instantaneous effects. However, Peters et al. (2013) also provide a special case in which the causal structure is identifiable with instantaneous effects. This special case occurs when the random variables of the model are jointly Gaussian, and the instantaneous effects are linear. Our framework extends to this special case.

[7]Recall that a distribution is faithful to a causal diagram if no conditional independence relations are present, other than the ones entailed by the Markov property.

since there is no reason to assume that faithfulness or causal minimality hold in practice. Theorem 1 by Gong et al. (2022) provides an identifiability result for a model with history-dependent noise and instantaneous effects. This result, however, requires causal minimality.

We remark that much of the previous related work considers the general problem of full causal discovery, whereas we use Axiom (A)-Axiom (C) for the more specific task of causal feature selection. However, our method can also be used for full causal discovery assuming a fully-observed acyclic system (see Section 5.2 for details).

### 3.2 Causal Structure

We are interested in direct causal effects, which are defined by distribution changes due to interventions on the DGP. An intervention amounts to actively manipulating the generative process of a potential cause $X^i$ at some time step $t$, without altering the remaining components of the DGP. Then, a time series $X^i$ has a direct effect on $Y$ if performing an intervention on some temporal variable $X_t^i$ will alter the distribution of $Y_T$, for some time steps $t, T$. For a precise definition of these concepts, we refer the reader to Appendix A.2.

Following, e.g., Mastakouri et al. (2021), we define the *full time* graph $\mathcal{G}$ as a directed graph whose edges represent all direct causal effects among the variables at all time steps. Given the outcome $Y_T$ at a given time step, we refer to the parent nodes in the full time graph as its *causal parents*. We further define the *summary* graph whose nodes are $X^i$ and $Y$, and with directed edges representing causal effects between the time series. We refer the reader to Figure 1 for a visualization of these graphs. Note that the causes of $Y$ correspond to the parent nodes of $Y$ in the summary graph.

### 3.3 Granger Causality

Granger causality (Granger, 1988) is one of the most commonly used approaches to infer causal relations from observational time-series data. Its central assumption is that "cause-effect relationships cannot work against time". Informally, if the prediction of the future of a target time-series $Y$ can be improved by knowing past elements of another time-series $X^i$, then $X^i$ "Granger causes" $Y$. Formally, we say that $X^i$ Granger causes $Y$ if it holds

$$\mathbb{P}\left(Y_T = y \mid X_T^i = x, I_T^{\backslash i} = i\right) \neq \mathbb{P}\left(Y_T = y \mid I_T^{\backslash i} = i\right) \tag{1}$$

for a non-zero probability event $\{X_T^i = x, I_T^{\backslash i} = i\}$, where $I_T$ stands for the set $\{X_T^1, X_T^2, \ldots, X_T^m\}$ and $I_T^{\backslash i}$ represents the set $I_T \setminus \{X_T^i\}$.

Granger causality is commonly used to identify causes. Assuming stationary, multivariate Granger causality analysis usually fits a vector autoregressive model to a given dataset. This model can be then used to determine the causes of a target $Y$. However, it is important to note that Granger causality does not imply true causality in general. This limitation was acknowledged by Granger himself in Granger (1988).

## 4 Methodology

### 4.1 Double Machine Learning (DoubleML)

DoubleML is a general framework for parameter estimation, which uses debiasing techniques to achieve $\sqrt{n}$-consistency (see, e.g., Rotnitzky et al. (2020); Chernozhukov et al. (2022)). In DoubleML, we consider the problem of estimating a parameter $\theta_0$ as a solution of an equation of the form $\mathbb{E}\left[\mathcal{L}(\theta_0, \eta_0)\right] = 0$. The score function $\mathcal{L}$ depends on two terms, the true parameter $\theta_0$ that we wish to estimate, and a nuisance parameter $\eta_0$. We do not directly care about the correctness of our estimate of $\eta_0$, as long as we get a good estimator of $\theta_0$. The nuisance parameter $\eta_0$ may induce an unwanted bias in the estimation process, resulting in slow convergence.

To overcome this problem, we consider score functions that fulfill the Mixed Bias Property (MBP) (Rotnitzky et al., 2020). Intuitively, the MBP holds if small changes of the nuisance parameter do not significantly affect the score function computed around the true parameters $(\theta_0, \eta_0)$. If the given score function fulfills the MBP, then it is possible to partially remove the bias induced by the nuisance parameter using (double) cross-fitting.[8] By removing this bias, the convergence rate of the parameter

---

[8] Although in our analysis we focus on the MBP, strong consistency results follow using the weaker Neyman Orthogonaloty condition.

estimation improves. We refer the reader to Rotnitzky et al. (2020); Chernozhukov et al. (2018); Rotnitzky et al. (2020); Chernozhukov et al. (2022) for details.

Following the notation introduced earlier, for a fixed time step $T$, let $\boldsymbol{X}$ be a random variable in the set of random variables $\boldsymbol{V}$, $g$ any real-valued function of $\boldsymbol{X}$ such that $\mathbb{E}\left[g^2(\boldsymbol{X})\right] < \infty$. We consider parameters of the form $\theta_0 := \mathbb{E}\left[m(\boldsymbol{V};g)\right]$, where $m(\boldsymbol{V};g)$ is a linear moment functional in $g$. The celebrated Riesz Representation Theorem ensures that, under certain conditions, there exists a function $\alpha_0$ of $\boldsymbol{X}$ such that $\mathbb{E}\left[m(\boldsymbol{V};g)\right] = \mathbb{E}\left[\alpha_0(\boldsymbol{X})g(\boldsymbol{X})\right]$. The function $\alpha_0$ is called the *Riesz Representer* (RR). Chernozhukov et al. (2021) shows that the Riesz representer can be estimated from samples. Using the RR, we can derive a score function for the parameter $\theta_0$ with $g_0(\boldsymbol{X}) = \mathbb{E}\left[Y \mid \boldsymbol{X}\right]$ that fulfills the MBP.

Chernozhukov et al. (2022); Rotnitzky et al. (2020) shows that we can learn the parameters $\theta_0$ as above, with a score function of the form

$$\varphi(\theta, \boldsymbol{\eta}) := m(\boldsymbol{V};g) + \alpha(\boldsymbol{X}) \cdot (Y - g(\boldsymbol{X})) - \theta. \tag{2}$$

Here, $\boldsymbol{\eta} := (\alpha, g)$ is a nuisance parameter consisting of a pair of square-integrable functions. As shown by Chernozhukov et al. (2022), the score function Eq. (2) yields $\mathbb{E}\left[\varphi(\theta_0, \boldsymbol{\eta})\right] = -\mathbb{E}\left[(\alpha(\boldsymbol{X}) - \alpha_0(\boldsymbol{X}))(g(\boldsymbol{X}) - g_0(\boldsymbol{X}))\right]$. This equation corresponds to the MBP as in Definition 1 of Rotnitzky et al. (2020). Hence, if we use (double) cross-fitting, the resulting estimator $\hat{\theta}$ has the *double robustness property*. That is, the quantity $\sqrt{n}(\hat{\theta} - \theta_0)$ converges in distribution to a zero-mean Normal distribution if the product of the mean-square convergence rates of $\alpha$ and $g$ is faster than $\sqrt{n}$.

## 4.2 GRANGER CAUSALITY IMPLIES TRUE CAUSATION UNDER AXIOM (A)-AXIOM (C)

As discussed in Section 3.3, Granger causality does not imply true causality in general. In our case, however, Granger causality corresponds to true causation, as stated in the following result.

**Theorem 4.1.** *Consider a causal model as in Axiom (A)-Axiom (C). Then, a time series $\boldsymbol{X}^i$ is a potential cause of $\boldsymbol{Y}$ if and only if (iff.) $\boldsymbol{X}^i$ Granger causes $\boldsymbol{Y}$.*

Proof of this result is given in Appendix A.5. Importantly, Theorem 4.1 does not require causal faithfulness.[7] We remark that Peters et al. (2013) shows that Granger causality implies true causation for fully autoregressive models (see also Löwe et al. (2022)). This result is based on the identifiability of additive noise models, in which all relevant variables are observed (Peters et al., 2011). Our framework, however, is more general than Peters et al. (2013); Löwe et al. (2022), since it allows for confounding among the covariates (see Figure 1). In the special case of a fully autoregressive model, Theorem 4.1 is equivalent to previous results (Peters et al., 2013; Löwe et al., 2022).

## 4.3 TESTING GRANGER CAUSALITY WITH DOUBLEML

Our approach to identifying potential causes consists of performing a statistical test to determine if Eq. 1 holds. Due to Theorem 4.1, a straightforward approach would then consist of using a conditional independence test, to select or discard a time series $\boldsymbol{X}^i$ as a cause of the outcome $\boldsymbol{Y}$. However, conditional independence testing is challenging in high-dimensional settings. Furthermore, kernel-based conditional independence tests (Fukumizu et al., 2007; Zhang et al., 2011; Park & Muandet, 2020; Sheng & Sriperumbudur, 2020) are computationally expensive. Instead, we provide a new statistical test based on DoubleML. Our approach is based on the observation that under Axiom (A)-Axiom (C), the condition in Eq. 1 can be written in terms of simple linear moment functionals. The following theorem holds.

**Theorem 4.2.** *Consider the notation as in Eq. 1, and fix a time step $T$. For any square-integrable random variable $g^0(\boldsymbol{X}_T^i, \boldsymbol{I}_T^{\setminus i})$, consider the moment functional $m_0(\boldsymbol{V};g^0) := Y_T \cdot g^0$. Similarly, for any square-integrable random variable $g^i(\boldsymbol{I}_T^{\setminus i})$, consider the moment functional $m_i(\boldsymbol{V};g^i) := Y_T \cdot g^i$. Assuming Axiom (A)-Axiom (D), $\boldsymbol{X}^i$ Granger causes $\boldsymbol{Y}$ iff. it holds*

$$\mathbb{E}\left[m_0(\boldsymbol{V};g_0^0)\right] - \mathbb{E}\left[m_i(\boldsymbol{V};g_0^i)\right] \neq 0, \tag{3}$$

*with $g_0^0(\boldsymbol{x}, \boldsymbol{i}) = \mathbb{E}[Y_T \mid \boldsymbol{X}_T^i = \boldsymbol{x}, \boldsymbol{I}_T^{\setminus i} = \boldsymbol{i}]$, and $g_0^i(\boldsymbol{i}) = \mathbb{E}[Y_T \mid \boldsymbol{I}_T^{\setminus i} = \boldsymbol{i}]$.*

---

**Algorithm 1** The SITD

---

1: split the dataset D into $k$ random disjoint subsets $D_j$ of the same size;

2: **for** each dataset partition $j$ **do**
3:     perform regressions to estimate $\alpha_j^0$, $g_j^0$ on dataset $D \setminus D_j$;
4:     $\theta_j^0 \leftarrow \hat{\mathbb{E}}_{D_j}[Y_T \cdot g_j^0 + \alpha_j^0 \cdot (Y_T - g_j^0)]$;
5: **end for**
6: $\theta^0 \leftarrow k^{-1} \sum_j \theta_j^0$;

7: **for** each potential cause $\boldsymbol{X}^i$ **do**
8:     **for** each dataset partition $j$ **do**
9:         apply zero-masking to $\alpha_j^0$, $g_j^0$ for the variables $\boldsymbol{X}^i$, and denote with $\tilde{\alpha}_j^0$, $\tilde{g}_j^0$ the resulting models;
10:         $\theta_j^i \leftarrow \hat{\mathbb{E}}_{D_j}[Y_T \cdot \tilde{g}_j^i + \tilde{\alpha}_j^i \cdot (Y_T - \tilde{g}_j^i)]$;
11:     **end for**
12:     $\theta^i \leftarrow k^{-1} \sum_j \theta_j^i$
13:     perform a paired Student's $t$-test to determine if $\theta^i \approx \theta^0$, and select time series $\boldsymbol{X}^i$ as a cause if the null-hypotheses is rejected;
14: **end for**

15: **return** the selected time series;

---

The proof is deferred to Appendix A.6. By this theorem, we can identify the causal parents of $Y$ by testing Eq. 3. This boils down to learning parameters $\theta_0^0 := \mathbb{E}[Y_T \cdot g_0^0]$, $\theta_0^i := \mathbb{E}[Y_T \cdot g_0^i]$, and then performing a sample test to verify if it holds $\theta^0 - \theta^i \neq 0$.

Intuitively, the parameter $\theta := \mathbb{E}[Y_T . \mathbb{E}[Y_T | I]]$ quantifies the cross-correlation between the target variable $Y_T$ and its conditional mean $\mathbb{E}[Y_T | I]$, given the set of variables $I$. In general terms, variations in the parameter $\theta$ due to changes in the set $I$ suggest the presence of a causal relationship between alterations in the set $I$ and the target variable $Y_T$.

**Outline of the approach.** Given a sample dataset D, we iterate the following steps:

Step (1) Select a potential cause $\boldsymbol{X}^i$ to test if $\boldsymbol{X}^i$ Granger causes $\boldsymbol{Y}$. Split the dataset D into $k$ disjoint sets $D_j$ with $k \geq 2$.

Step (2) Learn $g_0^0$, $g_0^i$ as in Theorem 4.2 on dataset $D \setminus D_j$. Denote with $g_j^0$, $g_j^i$ the resulting estimates. Learn the RRs $\alpha_0^0$, $\alpha_0^i$ coresponding to $\mathbb{E}[Y_T \cdot g_0^0]$ and $\mathbb{E}[Y_T \cdot g_0^i]$ respectively, on dataset $D \setminus D_j$. Denote with $\alpha_j^0$, $\alpha_j^i$ the resulting estimates.

Step (3) Learn the parameter $\theta_0^0 = \mathbb{E}[Y_T \cdot g_0^0]$ on dataset $D_j$ for all $j$, by minimizing the orthogonal score function as in Eq. 2. Learn the parameter $\theta_0^i = \mathbb{E}[Y_T \cdot g_0^i]$ on dataset $D_j$ for all $j$, by minimizing the orthogonal loss as in Eq. 2. Denote with $\theta_j^0$, $\theta_j^i$ the resulting estimates, and define $\theta^0 := k^{-1} \sum_j \theta_j^0$ and $\theta^i := k^{-1} \sum_j \theta_j^i$.

Step (4) Use a two-sample Student's $t$-test to determine if $\mathbb{E}[\theta^0 - \theta^i]$ is approximately zero. Select $\boldsymbol{X}^i$ as a cause of $\boldsymbol{Y}$ if the null hypothesis is rejected.

This procedure can be iterated for each of the features.

**Strong consistency guarantees.** Step (4) requires some explanation. Under mild conditions on the convergence of $g_j^0, g_j^i$ and $\alpha_j^0, \alpha_j^i$, the quantity $\theta^0 - \theta^i$ has $\sqrt{n}$-consistency. As a consequence of strong consistency, it holds

$$\lim_{n \to +\infty} \sqrt{n} \mathbb{E}[\theta^0 - \theta^i] = 0, \tag{4}$$

iff. $\mathbb{E}[Y_T \cdot \eta_0^0] - \mathbb{E}[Y_T \cdot \eta_0^i] = 0$. Then, by Theorem 4.2, Eq. 4 holds iff. $\boldsymbol{X}^i$ does not Granger causes $\boldsymbol{Y}$. Step (4) in the procedure outlined above uses a two-sample Student's $t$-test to determine if $\boldsymbol{X}^i$ Granger causes $\boldsymbol{Y}$. We refer the reader to Chernozhukov et al. (2022; 2018) for a proof of the $\sqrt{n}$-consistency for estimates as $\theta^0$ and $\theta^i$.

## 5 THE ALGORITHM

### 5.1 OVERVIEW

Our approach to test Granger causality essentially follows Step (1)-Step (4) outlined in the previous section. We refer to our approach as the SITD (Structure Identification from Temporal Data). Our method is presented in Algorithm 1, where we use the notation $\hat{\mathbb{E}}_D$ to denote the empirical expected value over a given dataset D. We note that, however, this procedure requires training $g_j^0$, $g_j^i$ and $\alpha_j^0$, $\alpha_j^i$ for each potential cause, and for each partition $D_j$. This can be time-consuming in large instances. To overcome this bottleneck, we make the following improvements:

- The parameters $g_j^0$, $\alpha_j^0$, $\theta_j^0$, and $\theta^0$ are the same for each one of the potential causes, and we only trained them once at the beginning of the process (Line 2-6 of Algorithm 1).
- The RR corresponding to $\mathbb{E}\left[m_0(\boldsymbol{V}; g_0^0)\right]$ as in Theorem 4.1 can be written in explicit form as $\alpha_0^0 = \mathbb{E}[Y_T \mid \boldsymbol{X}_T^i = \boldsymbol{x}, \boldsymbol{I}_T^{\setminus i} = \boldsymbol{i}]$. Similarly, the RR of $\mathbb{E}\left[m_i(\boldsymbol{V}; g_i^0)\right]$ as in Theorem 4.1 can be written in explicit form as $\alpha_0^i = \mathbb{E}[Y_T \mid \boldsymbol{I}_T^{\setminus i} = \boldsymbol{i}]$. Hence, we estimate $\alpha_0^0$, $\alpha_0^i$ by performing regressions.
- Instead of computing $g_0^i$ and $\alpha_0^i$ directly, we apply a zero-masking layer to $g_j^0$ and $\alpha_j^0$ for the features $\boldsymbol{X}^i$. This masking layer tells the sequence-processing layers that the input values for features $\boldsymbol{X}^i$ should be skipped. We then compute $\theta_0^i$ using the resulting surrogate models $\tilde{g}_j^i$, $\tilde{\alpha}_j^i$ (Line 9-10 in Algorithm 1). Please refer to Appendix A.7 on why zero-masking is not hurting the estimations.

**Runtime of Algorithm 1.** Much of the run time of our algorithm consists of performing a regression to learn $\eta_j^0$ on dataset $D_j$. Denote with $d$ the time complexity of performing such a regression. For a given $k$-partition of the dataset, we can upper-bound the time complexity of our algorithm as $\mathcal{O}(dk)$. Here, $d$ depends on the specific techniques used for the regression. Non-parametric regression can be performed efficiently in the problem size. We analyse the computational complexity in Appendix A.9 further and showp runtime plots for extensive experiments in Appendix A.10.3.

### 5.2 FULL CAUSAL DISCOVERY

It is possible to use Algorithm 1 for full causal discovery, for fully-observed acyclic auto-regressive models with no instantaneous effects (see Peters et al. (2013); Löwe et al. (2022) for a precise definition of this restricted framework). In fact, under these more restrictive assumptions, we can identify the causes of each random variable of the model by testing Granger causality. For these models, we can learn the full summary graph, by identifying the causes of each variable with Algorithm 1. The resulting run time can be quantified as $\mathcal{O}(mdk)$, where $d$ is the time complexity of performing a regression, as outlined above, $m$ is the number of time series considered, and $k$ is the number of dataset partitions used for double cross-fitting. Since $k = \mathcal{O}(1)$ w.r.t. parameters of the problem (see Appendix A.8), the runtime of SITD is $\mathcal{O}(md)$.

## 6 EXPERIMENTS

### 6.1 SEMI-SYNTHETIC EXPERIMENTS

**The Dream3 dataset.** Following previous related work (Tank et al., 2018; Khanna & Tan, 2019; Nauta et al., 2019; Bussmann et al., 2021; Gong et al., 2022), we evaluate performance with the Dream3 benchmark (Prill et al., 2010; Marbach et al., 2009). Dream3 consists of measurements of gene expression levels for five different networks. Each network contains 100 genes. In this dataset, time series correspond to different perturbation trajectories. For each one of the five networks, there are $m = 46$ different trajectories.

**Description of the experiments.** Our goal is to infer the causal structure of each group of time series (trajectories) of Dream3, as detailed above. This gives us a total of five tasks, i.e., E.Coli 1, E.Coli 2, Yeast 1, Yeast 2, and Yeast 3. We run our algorithm on this dataset and we use the area under the ROC curve (AUROC) as the performance metric. Following (Gong et al., 2022), we compare against the following baselines: cMLP (Tank et al., 2018), cLSTM (Tank et al., 2018), TCDF (Nauta et al., 2019), SRU (Khanna & Tan, 2019), eSRU (Khanna & Tan, 2019), Dynotears (Pamfil et al., 2020), Rhino+g (Gong et al., 2022), and Rhino (Gong et al., 2022).

Table 1: AUROC score for the SITD and common algorithms for Granger causality. Models with "✦" sign use deep neural networks. We remark that our estimator is much simpler than deep nets, and it has low sample complexity. We observe that our method achieves state-of-the-art performance on three tasks (E.Coli 1, E.Coli 2, Yeast 2) and that it obtains comparable performance on the remaining two (Yeast 1, Yeast 3).

| Method | E.Coli 1 | E.Coli 2 | Yeast 1 | Yeast 2 | Yeast 3 |
|---|---|---|---|---|---|
| cMLP✦ | 0.644 | 0.568 | 0.585 | 0.506 | 0.528 |
| cLSTM✦ | 0.629 | 0.609 | 0.579 | 0.519 | 0.555 |
| TCDF✦ | 0.614 | 0.647 | 0.581 | 0.556 | 0.557 |
| SRU✦ | 0.657 | 0.666 | 0.617 | 0.575 | 0.55 |
| eSRU✦ | 0.66 | 0.629 | 0.627 | 0.557 | 0.55 |
| DYNO. | 0.590 | 0.547 | 0.527 | 0.526 | 0.510 |
| PCMCI$^+$ | 0.530 | 0.519 | 0.530 | 0.510 | 0.512 |
| Rhino✦ Reprod. | 0.686 | 0.625 | **0.667** | 0.580 | 0.540 |
| Rhino+g✦ Reprod. | 0.662 | 0.606 | 0.663 | **0.585** | **0.577** |
| **SITD (ours)** | **0.704**±0.005 | **0.680**±0.004 | 0.653±0.001 | **0.585**±0.003 | 0.544±0.003 |

**Results.** The results for cMLP, cLSTM, TCDF, SRU and SRU are taken directly from Khanna & Tan (2019); Gong et al. (2022). Regarding Rhino+g and Rhino, we partially reproduce the experiments by Gong et al. (2022), using their source code. Our implementation of Rhino+g and Rhino differs from Gong et al. (2022) only in the choice for the hyper-parameters, which is the same on all five tasks. We specifically use the following hyper-parameters for Rhino: Node Embedd. = 16, Instantaneous eff. = False, Node Embedd. (flow) = 16, lag = 2, $\lambda_s = 19$, Auglag = 30. And we use the following for Rhino+g: Node Embedd. = 16, Instantaneous eff. = False, lag = 2, $\lambda_s = 15$, Auglag = 60. This is the setting that is reported for the Ecoli1 subtask and found in the corresponding code implementation. We opted for this approach because in the experiment by Gong et al. (2022), it seems that Rhino overfitted to the dataset. We learn $\eta_j^i$ as in Line 3 of Algorithm 1, using a simple kernel ridge regression model with polynomial kernels of degree three combined with zero masking.

Overall, we observe that our method gives competitive results. Specifically, SITD outperforms all the other benchmarks on two tasks (E.Coli 1, E.Coli 2), and it obtains the same AUROC as Rhino+g on Yeast 2. Furthermore, SITD obtains comparable performance on the remaining tasks (Yeast 1 and Yeast 3). We also would like to emphasize that our estimator is much simpler than deep nets such as Rhino or Rhino+g. As such, it has lower sample complexity and lower run time than the other algorithms.

## 6.2 Additional experiments

We provide additional extensive synthetic experiments for various noise-to-signal ratios and number of causes in Appendix A.10. One appealing property of SITD is the double robustness property. Due to this property, the dependence of SITD on the estimator is relatively lower compared to existing baselines. Using a fast and simple estimator with low statistical complexity in Algorithm 1, we are able to get competitive results compared to richer models especially in low sample complexity settings. The illustrated case with a simple kernel ridge regression estimator with polynomial kernels of degree three in Section 6.1 is a compelling example of this. In Appendix A.10, we will thus focus on these practically important aspects in more detail and show in particular further experiments on low sample settings in Appendix A.10.2 and runtime plots in Appendix A.10.3.

## 7 Discussion

In this work, we propose an efficient algorithm for doubly robust structure identification from temporal data. We further provide asymptotical guarantees that our method is able to discover the direct causes even when there are cycles or hidden confounding and that our algorithm has $\sqrt{n}$-consistency. We extensively discuss the relations of the approach between the popular frameworks of Granger and Pearl's causality as well as relate and extend approaches from debiased machine learning to structure discovery from temporal data. We hope to apply these approaches to important real-world applications in biomedicine where robustness to confounding, sample efficiency, and ease of use are important for causal discovery from observational time series.

REPRODUCIBILITY STATEMENT

To ensure the reproducibility of our findings, we conducted extensive benchmarking experiments on synthetic datasets that provide a known ground truth. Detailed descriptions of the dataset generation process and the experiments can be found in Appendix A.10. Furthermore, we provide detailed proofs in Appendix A.5 and A.6 for the theoretical results as well as intuitions for zero-masking in Appendix A.7. Finally, wee have included the code for dataset generation and experimentation in the supplementary materials, and upon acceptance of the paper, we are committed to releasing it under an open-source license. This transparency enables fellow researchers to replicate our work effectively.

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

# A  APPENDIX

## A.1  BROADER IMPACT AND LIMITATIONS

In many safety-critical applications, one needs to infer causal structure from observational data e.g. in healthcare, economics, or geosciences since interventions are unethical or simply impossible. For these cases, one requires algorithms that have theoretical guarantees, are robust to hidden confounding, and are computationally efficient. We hope that in the long term, our work can thus contribute to better machine learning-driven decision-making in safety-critical environments.

While theoretical guarantees and extensive experiments are critical components for the evaluation of algorithms, especially in safety-critical environments this can potentially lead to a false sense of security and trust. Constant monitoring and assessment by domain experts are needed before and during the deployment of any machine learning algorithm, especially in safety-critical environments.

One key limitation of our approach as well as of all causal discovery approaches from observational data is the reliance on additional assumptions. These assumptions are required since causal discovery from observational data is impossible without them. We related our assumptions to previous ones in the Framework Section 3. These assumptions of our approach need to be checked before applying the approach but are less restrictive than for comparable baselines. Moreover, since our approach is inspired by debiased machine learning, the focus is on inferring the true or "debiased" underlying structure rather than obtaining low variance estimates of the structure.

## A.2  DIRECT CAUSAL EFFECTS AND INTERVENTIONS

In this section, we clarify the notion of an intervention and direct causal effects. We will also introduce some notation that will be used later on in the proofs.

We consider interventions by which a random variable $X_t^j$ is set to a constant $X_t^j \leftarrow x$. We denote with $Y_T \mid do(X_t^i = x)$ the outcome time series $\boldsymbol{Y}$ at time step $T$, after performing an intervention as described above. We can likewise perform multiple joint interventions, by setting a group of random variables $\boldsymbol{I}$ at different time steps, to pre-determined constants specified by an array $\boldsymbol{i}$. We use the symbol $Y_T \mid do(\boldsymbol{I} = \boldsymbol{i})$ to denote the resulting post-interventional outcome, and we denote with $\mathbb{P}\left(Y_T = y \mid do(\boldsymbol{I} = \boldsymbol{i})\right)$ the probability of the event $\{Y_T \mid do(\boldsymbol{I} = \boldsymbol{i}) = y\}$.

Using this notation, a time series $\boldsymbol{X}^i$ has a direct effect on the outcome $\boldsymbol{Y}$, if performing different interventions on the variables $\boldsymbol{X}^i$, while keeping the remaining variables fixed, will alter the probability distribution of the outcome $\boldsymbol{Y}$. Formally, define the sets of random variables $\boldsymbol{I}_T := \{X_t^1, \ldots, X_t^n, Y_t\}_{t<T}$, which consists of all the information before time step $T$. Similarly, define the random variable $\boldsymbol{X}_T^i := \{X_t^i\}_{t<T}$, consisting of all the information of time series $\boldsymbol{X}^i$ before time step $T$. Define the variable $\boldsymbol{I}_T^{\backslash i} := \boldsymbol{I}_T \setminus \boldsymbol{X}_T^i$, which consists of all the variables in $\boldsymbol{I}_T$ except for $\boldsymbol{X}_T^i$. Then, a time series $\boldsymbol{X}^i$ has a direct effect on the outcome $\boldsymbol{Y}$ if it holds

$$\mathbb{P}\left(Y_T = y \mid do\left(\boldsymbol{X}_T^i = \boldsymbol{x}', \boldsymbol{I}_T^{\backslash i} = \boldsymbol{i}\right)\right) \neq \mathbb{P}\left(Y_T = y \mid do\left(\boldsymbol{X}_T^i = \boldsymbol{x}'', \boldsymbol{I}_T^{\backslash i} = \boldsymbol{i}\right)\right) \qquad (5)$$

We say that a time series $\boldsymbol{X}^i$ causes $\boldsymbol{Y}$, if there is a direct effect between $\boldsymbol{X}^i$ and $\boldsymbol{Y}$ as in Eq. 5, for any time step $T$.

## A.3  NECESSITY OF THE STATISTICAL INDEPENDENCE OF $\varepsilon_t$

We provide a counterexample, to show that if there are dependencies between the noise and the historical data, then the causal structure may not be identifiable from observational data. To this end, we consider a first dataset $\{X_t, Y_t\}_{t \in \mathbb{Z}}$, defined as

$$X_{t-1}, Y_t \sim \mathcal{N}(\boldsymbol{0}, \Sigma)$$

Here, $\mathcal{N}(\boldsymbol{0}, \Sigma)$ is a zero-mean joint Gaussian distribution with covariance matrix

$$\Sigma = \begin{bmatrix} 1 & 1 \\ 1 & 1 \end{bmatrix}.$$

We also consider a second dataset $\{W_t, Z_t\}_{t \in \mathbb{Z}}$, defined as

$$W_t \sim \mathcal{N}(0, 1), \quad Z_t = W_{t-1}$$

The parameter $\Sigma$ is defined as above. Both datasets entail the same joint probability distribution. However, the causal diagrams change from one dataset to the other. Hence, the causal structure cannot be recovered from observational data, if the posterior additive noise $\varepsilon_t$ is correlated with some of the covariates.

### A.4 Necessity of No Instantaneous causal effect between $Y$ and any of the potential causes $X^i$

Here, we provide a counterexample to show that without the no instantaneous causal effect, the causal structure may not be identifiable from observational data. Consider the following two models:

- Model 1: We consider time series $\{X_t\}$, $\{Y_t\}$ of the form $X_t = \mathbb{E}[X_{t-1}] + c$ and $Y_t = \mathbb{E}[Y_{t-1} - X_{t-1}] + X_t$. In this model, $c$ is a random variable drawn from a Gaussian distribution with a mean of 0 and covariance of 1.

- Model 2: We consider time series $\{X_t\}$, $\{Y_t\}$ of the form $Y_t = \mathbb{E}[Y_{t-1}] + c$ and $X_t = \mathbb{E}[X_{t-1} - Y_{t-1}] + Y_t$. In this model, $c$ is a random variable drawn from a Gaussian distribution with a mean of 0 and covariance of 1.

Both models entail the same joint distribution. However, in Model 1 $X$ has a causal effect on $Y$, whereas in Model 2 $Y$ has a causal effect on $X$. Hence, in this example, the causal structure is not identifiable.

### A.5 Proof of Theorem 4.1

We prove the following result.

**Theorem 4.1.** *Consider a causal model as in Axiom (A)-Axiom (C). Then, a time series $X^i$ is a potential cause of $Y$ if and only if (iff.) $X^i$ Granger causes $Y$.*

*Proof.* We first prove that it holds

$$\mathbb{P}\left(Y_T = y \mid do(X_t^i = x, \boldsymbol{I}_T^{\backslash i} = \boldsymbol{i})\right) = \mathbb{P}\left(Y_T = y \mid X_t^i = x, \boldsymbol{I}_T^{\backslash i} = \boldsymbol{i}\right) \tag{6}$$

for any non-zero event $\{Y_T = y\}$. To this end, define the group $\boldsymbol{P}$ consisting of all the causal parents of the outcome. Note that $\boldsymbol{P} \subseteq \{X_t^i, \boldsymbol{I}_T^{\backslash i}\}$. By Axiom (A), the outcome can be described as $Y = f(\boldsymbol{P}) + \varepsilon$, where $\varepsilon$ is independent of $\{X_t^i, \boldsymbol{I}_T^{\backslash i}\}$. Hence by Rule 2 of the do-calculus (see Pearl (2000), page 85) Eq. 6 holds, since $Y$ becomes independent of $\{X_t^i, \boldsymbol{I}_T^{\backslash i}\}$ once all arrows from $\boldsymbol{P}$ to $Y$ are removed from the graph of the DGP.

We now prove the claim using Eq. 6. To this end, assume that Eq. 6 holds and suppose that $X^i$ does not Granger causes $Y$, i.e., it holds

$$\mathbb{P}\left(Y_T = y \mid X_t^i = x, \boldsymbol{I}_T^{\backslash i} = \boldsymbol{i}\right) = \mathbb{P}\left(Y_T = y \mid \boldsymbol{I}_T^{\backslash i} = \boldsymbol{i}\right), \tag{7}$$

for any non-zero event $\{Y_T = y\}$. Then,

$$\mathbb{P}\left(Y_T = y \mid do(X_t^i = x, \boldsymbol{I}_T^{\backslash i} = \boldsymbol{i})\right) = \mathbb{P}\left(Y_T = y \mid X_t^i = x, \boldsymbol{I}_T^{\backslash i} = \boldsymbol{i}\right) \qquad \text{[Eq. 6]}$$

$$= \mathbb{P}\left(Y_T = y \mid \boldsymbol{I}_T^{\backslash i} = \boldsymbol{i}\right) \qquad \text{[Eq. 7]}$$

$$= \mathbb{P}\left(Y_T = y \mid X_t^i = x', \boldsymbol{I}_T^{\backslash i} = \boldsymbol{i}\right) \qquad \text{[Eq. 7]}$$

$$= \mathbb{P}\left(Y_T = y \mid do(X_t^i = x', \boldsymbol{I}_T^{\backslash i} = \boldsymbol{i})\right). \qquad \text{[Eq. 6]}$$

Hence, causality implies Granger causality.

We now prove that Granger causality implies causality. To this end, suppose that $X^i$ is not a potential cause of $Y$. By the definition of direct effects, it holds

$$\mathbb{P}\left(Y_T = y \mid do(X_t^i = x, \boldsymbol{I}_T^{\backslash i} = \boldsymbol{i})\right)$$

$$= \mathbb{E}\left[\mathbb{P}\left(Y_T = y \mid do(X_t^i = x', \boldsymbol{I}_T^{\backslash i} = \boldsymbol{i})\right) \mid \boldsymbol{I}_T^{\backslash i} = \boldsymbol{i}\right]. \tag{8}$$

Hence,

$$\mathbb{P}\left(Y_T = y \mid do(X_t^i = x, \boldsymbol{I}_T^{\setminus i} = \boldsymbol{i})\right)$$

$$= \mathbb{P}\left(Y_T = y \mid X_t^i = x, \boldsymbol{I}_T^{\setminus i} = \boldsymbol{i}\right) \qquad\qquad \text{[Eq. 6]}$$

$$= \mathbb{E}\left[\mathbb{P}\left(Y_T = y \mid do(X_t^i, \boldsymbol{I}_T^{\setminus i})\right) \mid \boldsymbol{I}_T^{\setminus i} = \boldsymbol{i}\right] \qquad \text{[Eq. 8]}$$

$$= \mathbb{E}\left[\mathbb{P}\left(Y_T = y \mid X_t^i, \boldsymbol{I}_T^{\setminus i}\right) \mid \boldsymbol{I}_T^{\setminus i} = \boldsymbol{i}\right] \qquad \text{[Eq. 6]}$$

$$= \mathbb{P}\left(Y_T = y \mid \boldsymbol{I}_T^{\setminus i} = \boldsymbol{i}\right),$$

and the claim follows. □

### A.6 MISSING PROOF OF 4.2

We prove the following result.

**Theorem 4.2.** *Consider the notation as in Eq. 1, and fix a time step $T$. For any square-integrable random variable $g^0(\boldsymbol{X}_T^i, \boldsymbol{I}_T^{\setminus i})$, consider the moment functional $m_0(\boldsymbol{V}; g^0) \coloneqq Y_T \cdot g^0$. Similarly, for any square-integrable random variable $g^i(\boldsymbol{I}_T^{\setminus i})$, consider the moment functional $m_i(\boldsymbol{V}; g^i) \coloneqq Y_T \cdot g^i$. Assuming Axiom (A)-Axiom (D), $\boldsymbol{X}^i$ Granger causes $\boldsymbol{Y}$ iff. it holds*

$$\mathbb{E}\left[m_0(\boldsymbol{V}; g_0^0)\right] - \mathbb{E}\left[m_i(\boldsymbol{V}; g_0^i)\right] \neq 0, \tag{3}$$

*with $g_0^0(\boldsymbol{x}, \boldsymbol{i}) = \mathbb{E}[Y_T \mid \boldsymbol{X}_T^i = \boldsymbol{x}, \boldsymbol{I}_T^{\setminus i} = \boldsymbol{i}]$, and $g_0^i(\boldsymbol{i}) = \mathbb{E}[Y_T \mid \boldsymbol{I}_T^{\setminus i} = \boldsymbol{i}]$.*

In order to prove Theorem 4.2, we use the following auxiliary lemma.

**Lemma A.1.** *Consider a causal model as in Axiom (A)-Axiom (C). Then, the following conditions are equivalent:*

1. $\mathbb{E}\left[Y_T \mid X_t^i = x, \boldsymbol{I}_T^{\setminus i} = \boldsymbol{i}\right] = \mathbb{E}\left[Y_T \mid X_t^i = x', \boldsymbol{I}_T^{\setminus i} = \boldsymbol{i}\right]$ *a.s.* ;

2. $\mathbb{P}\left(Y_T = y \mid X_t^i = x, \boldsymbol{I}_T^{\setminus i} = \boldsymbol{i}\right) = \mathbb{P}\left(Y_T = y \mid X_t^i = x', \boldsymbol{I}_T^{\setminus i} = \boldsymbol{i}\right)$ *a.s.*

*Proof.* Clearly, Item 2 implies Item 1.

We now prove the converse, i.e., we show that Item 1 implies Item 2. To this end, define the group $\boldsymbol{P}_T$ consisting of all the causal parents of $Y_T$. Note that it holds $\boldsymbol{P}_T \subseteq \{\boldsymbol{I}_T^{\setminus i}, X_t^i\} \subseteq \{\boldsymbol{I}_T^{\setminus i}, X_t^i\}$. Hence, the joint intervention $\{X_t^i, \boldsymbol{i}_{T-1}^{i,t}\} \leftarrow \{x, \boldsymbol{i}\}$ define an intervention on the parents $\boldsymbol{P}_T \leftarrow \boldsymbol{p}$. Further, we can write the potential outcome as

$$Y_T \mid do(X_t^i = x, \boldsymbol{I}_T^{\setminus i} = \boldsymbol{i}) = f(\boldsymbol{p}) + \varepsilon. \tag{9}$$

Similarly, the joint intervention $\{X_t^i, \boldsymbol{I}_{T-1}^{i,t}\} \leftarrow \{x, \boldsymbol{i}\}$, define an intervention on the parents $\boldsymbol{P}_T \leftarrow \boldsymbol{p}'$. We can write the potential outcome as

$$Y_T \mid do(X_t^i = x', \boldsymbol{I}_T^{\setminus i} = \boldsymbol{i}) = f(\boldsymbol{p}') + \varepsilon. \tag{10}$$

Hence, it holds

$$f(\boldsymbol{p}) + \mathbb{E}\left[\varepsilon\right] = \mathbb{E}\left[Y_T \mid do(X_t^i = x, \boldsymbol{I}_T^{\setminus i} = \boldsymbol{i})\right] \qquad \text{[Eq. 9]}$$

$$= \mathbb{E}\left[Y_T \mid X_t^i = x, \boldsymbol{I}_T^{\setminus i} = \boldsymbol{i}\right] \qquad \text{[Eq. 6, Theorem 4.1]}$$

$$= \mathbb{E}\left[Y_T \mid X_t^i = x', \boldsymbol{I}_T^{\setminus i} = \boldsymbol{i}\right] \qquad \text{[by assumption]}$$

$$= \mathbb{E}\left[Y_T \mid do(X_t^i = x', \boldsymbol{I}_T^{\setminus i} = \boldsymbol{i})\right] \qquad \text{[Eq. 6, Theorem 4.1]}$$

$$= f(\boldsymbol{p}') + \mathbb{E}\left[\varepsilon\right]. \qquad \text{[Eq. 10]}$$

By Axiom (A), the variable $\varepsilon$ is exogenous independent noise. From the chain of equations above it follows that $f(\boldsymbol{p}) = f(\boldsymbol{p}')$. Hence,

$$\mathbb{P}\left(Y_T = y \mid do(X_t^i = x, \boldsymbol{I}_T^{\backslash i} = \boldsymbol{i})\right) = \mathbb{P}\left(f(\boldsymbol{p}) + \varepsilon = y\right)$$

$$= \mathbb{P}\left(f(\boldsymbol{p}') + \varepsilon = y\right) = \mathbb{P}\left(Y_T = y \mid do(X_t^i = x', \boldsymbol{I}_T^{\backslash i} = \boldsymbol{i})\right) \qquad (11)$$

We conclude that it holds

$$\mathbb{P}\left(Y_T = y \mid X_t^i = x, \boldsymbol{I}_T^{\backslash i} = \boldsymbol{i}\right)$$

$$= \mathbb{P}\left(Y_T \mid do(X_t^i = x, \boldsymbol{I}_T^{\backslash i} = \boldsymbol{i})\right) \qquad \text{[Eq. 6, Theorem 4.1]}$$

$$= \mathbb{P}\left(Y_T = y \mid do(X_t^i = x', \boldsymbol{I}_T^{\backslash i} = \boldsymbol{i})\right) \qquad \text{[Eq. 11]}$$

$$= \mathbb{P}\left(Y_T = y \mid X_t^i = x', \boldsymbol{I}_T^{\backslash i} = \boldsymbol{i}\right), \qquad \text{[Eq. 6, Theorem 4.1]}$$

as claimed. $\qquad\qquad\square$

We can now prove the main result.

*Proof of Theorem 4.2.* We first prove that $X^i$ Granger causes $Y$ iff. it holds

$$\mathbb{E}\left[\left(\mathbb{E}\left[Y_T \mid X_t^i, \boldsymbol{I}_T^{\backslash i}\right] - \mathbb{E}\left[Y_T \mid \boldsymbol{I}_T^{\backslash i}\right]\right)^2\right] \neq 0. \qquad (12)$$

First, suppose that Eq. 12 does not hold. Then, it holds $[Y_T \mid X_t^i = x, \boldsymbol{I}_T^{\backslash i} = \boldsymbol{i}] = \mathbb{E}[Y_T \mid X_t^i = x', \boldsymbol{I}_T^{\backslash i} = \boldsymbol{i}]$, a.s. Combining this equation with Lemma A.1 yields

$$\mathbb{P}\left(Y_T = y \mid X_t^i = x, \boldsymbol{I}_T^{\backslash i} = \boldsymbol{i}\right) = \mathbb{E}\left[\mathbb{P}\left(Y_T = y \mid X_t^i, \boldsymbol{I}_T^{\backslash i}\right) \mid \boldsymbol{I}_T^{\backslash i} = \boldsymbol{i}\right]$$

$$= \mathbb{P}\left(Y_T = y \mid \boldsymbol{I}_T^{\backslash i} = \boldsymbol{i}\right),$$

a.s. Hence, if $X^i$ Granger causes $Y$, then Eq. 12 holds.

$$\mathbb{E}\left[Y_T \mid X_t^i = x, \boldsymbol{I}_T^{\backslash i} = \boldsymbol{i}\right] \neq \mathbb{E}\left[Y_T \mid X_t^i = x', \boldsymbol{I}_T^{\backslash i} = \boldsymbol{i}\right], \qquad (13)$$

for a triple $\{x, x', \mathbf{w}\}$. By combining Eq. 13 with Lemma A.1 we see that Eq. 12 implies causality. However, by Theorem 4.1 Granger causality is equivalent to causality in this case.

We now prove the claim. By the tower property of the expectation Williams (1991) that

$$\mathbb{E}\left[\left(\mathbb{E}\left[Y_T \mid X_t^i, \boldsymbol{I}_T^{\backslash i}\right] - \mathbb{E}\left[Y_T \mid \boldsymbol{I}_T^{\backslash i}\right]\right)^2\right]$$

$$= \mathbb{E}\left[\left(\mathbb{E}\left[Y_T \mid X_t^i, \boldsymbol{I}_T^{\backslash i}\right] - \mathbb{E}\left[Y_T \mid \boldsymbol{I}_T^{\backslash i}\right]\right)^2\right]$$

$$= \mathbb{E}\left[\mathbb{E}\left[\left(\mathbb{E}\left[Y_T \mid X_t^i, \boldsymbol{I}_T^{\backslash i}\right] - \mathbb{E}\left[Y_T \mid \boldsymbol{I}_T^{\backslash i}\right]\right)^2 \mid \boldsymbol{I}_T^{\backslash i}\right]\right]$$

$$= \mathbb{E}\left[\mathbb{E}\left[\left(\mathbb{E}\left[Y_T \mid X_t^i, \boldsymbol{I}_T^{\backslash i}\right]^2 - \mathbb{E}\left[Y_T \mid X_t^i, \boldsymbol{I}_T^{\backslash i}\right] \mathbb{E}\left[Y_T \mid \boldsymbol{I}_T^{\backslash i}\right]\right) \mid \boldsymbol{I}_T^{\backslash i}\right]\right]$$

$$= \mathbb{E}\left[\mathbb{E}\left[Y_T \mid X_t^i, \boldsymbol{I}_T^{\backslash i}\right]^2\right] - \mathbb{E}\left[\mathbb{E}\left[\left(\mathbb{E}\left[Y_T \mid X_t^i, \boldsymbol{I}_T^{\backslash i}\right] \mathbb{E}\left[Y_T \mid \boldsymbol{I}_T^{\backslash i}\right]\right) \mid \boldsymbol{I}_T^{\backslash i}\right]\right]$$

$$= \mathbb{E}\left[\mathbb{E}\left[Y_T \mid X_t^i, \boldsymbol{I}_T^{\backslash i}\right]^2\right] - \mathbb{E}\left[\mathbb{E}\left[Y_T \mid \boldsymbol{I}_T^{\backslash i}\right]^2\right]$$

$$= \mathbb{E}\left[Y_T \mathbb{E}\left[Y_T \mid X_t^i, \boldsymbol{I}_T^{\backslash i}\right]\right] - \mathbb{E}\left[Y_T \mathbb{E}\left[Y_T \mid \boldsymbol{I}_T^{\backslash i}\right]\right],$$

as claimed. $\qquad\qquad\square$

## A.7 INTUITION ON ZERO-MASKING

We provide intuition why masking is a reasonably good idea. Assume that the function $\widehat{f}$ is a $\epsilon$-close estimator of the true function $f^*$ in the $\mathcal{L}^2(P_{X_1, X_2, \ldots, X_m})$ norm, where the functions $\widehat{f}, f^*$ and the joint probability distribution $P_{X_2, \ldots, X_m}$ are defined on the set $\{X_1, X_2, \ldots, X_m\}$,

$$||\widehat{f} - f^*||_{\mathcal{L}^2(P_{X_1, X_2, \ldots, X_m})} \leq \epsilon$$

Now, let's mask the random variable $X_1$. We are interested to see how close is the estimator $\mathbb{E}_{X_1} \widehat{f}$ to the true function $\mathbb{E}_{X_1} f^*$ in the $\mathcal{L}^2(P_{X_2, \ldots, X_m})$ norm, where the functions $\mathbb{E}_{X_1} \widehat{f}, \mathbb{E}_{X_1} f^*$ and the marginal probability distribution $P_{X_2, \ldots, X_m}$ are defined on the rest of variables $\{X_2, \ldots, X_m\}$, By Jensen's inequality, we infer that for any realization of $X_2 = x_2, X_3 = x_3, \ldots, X_m = x_m$,

$$(\mathbb{E}_{X_1} \widehat{f}(X_1, x_2, \ldots, x_m) - \mathbb{E}_{X_1} f^*(X_1, x_2, \ldots, x_m))^2 \leq \mathbb{E}_{X_1}[(\widehat{f}(X_1, x_2, \ldots, x_m) - f^*(X_1, x_2, \ldots, x_m))^2]$$

Plugging it in the $\epsilon$-closeness assumption leads to,

$$||\mathbb{E}_{X_1} \widehat{f} - \mathbb{E}_{X_1} f^*||_{\mathcal{L}^2(P_{X_2, \ldots, X_m})} \leq ||\widehat{f} - f^*||_{\mathcal{L}^2(P_{X_1, X_2, \ldots, X_m})} \leq \epsilon,$$

which guarantees that $\mathbb{E}_{X_1} \widehat{f}$ is also $\epsilon$-closs to $\mathbb{E}_{X_1} f^*$ and hence it's a good estimator. In the sequel, a natural solution would be to estimate $\mathbb{E}_{X_1} \widehat{f}$ by taking averages of $\widehat{f}$ over different samples of $X_1$. However, for the linear regression problem that the estimator has a linear structure of the input, it is straightforward to show that it is enough to evaluate $\widehat{f}$ at $\mathbb{E}[X_1]$. And finally due to the zero-centering step of data preprocessing, $\mathbb{E}[X_1] = 0$. Thus, the aforementioned procedure is equivalent to zero-masking.

## A.8 A NOTE ON THE NUMBER OF PARTITIONS

The number of partitions $k$ affects the performance of our algorithm in practice since a larger number of partitions will help in removing a bias in the estimates. However, in our experiments, we observe that a small number of partitions is sufficient to achieve good results. Furthermore, an excessive number of random partitions may have a detrimental effect, due to the possible small number of samples in each partition. Hence, we believe that the number of partitions will not drastically affect performance in practice. Reasonable choices of $k$ for our experiments range between 3-7, hence $k = \mathcal{O}(1)$ w.r.t. parameters of the problem. Thus, the resulting runtime can be reported as $\mathcal{O}(md)$.

## A.9 COMPUTATIONAL COMPLEXITY AND COMPARISON

As discussed in Section 2, compared to SITD, conditional independence-based approaches such as PCMCI (Runge et al., 2019b), PCMCI+ (Runge, 2020), and LPCMCI (Gerhardus & Runge, 2020) face exponential computational barriers. It is widely known that even endowed with a perfect infinite sample independence testing oracle, learning Bayesian Networks becomes NP-Hard (Chickering et al., 2004; Chickering, 1996). Consequently, computational challenges arise not only due to the nature of the conditional independence tests themselves but also from the computational intractability of searching through the exponentially large space of possible network structures. Hence, the runtime of the $\mathcal{O}(m)$ number of regressions that SITD demands is negligible compared to the exponential number of conditional independence tests from lengthy time-series. To support this argument in practice, we provide a runtime comparison between SITD and PCMCI+ w.r.t. the number of nodes $m$ in Table 2.

|        | 10        | 20          | 30         | 40         | 50            | 100       | 200        | 400          |
|--------|-----------|-------------|------------|------------|---------------|-----------|------------|--------------|
| SITD   | $42 \pm 13$ | $35 \pm 6$    | $29 \pm 1$   | $30 \pm 4$   | $35 \pm 15$     | $41 \pm 6$  | $62 \pm 6$   | $77 \pm 10$    |
| PCMCI+ | $4.2 \pm 0.4$ | $18.6 \pm 0.8$ | $48.6 \pm 1.4$ | $99.2 \pm 10$ | $216.4 \pm 42.2$ | $1091 \pm 65$ | $5678 \pm 264$ | $\approx 8$ hours |

Table 2: Table with runtime means and standard deviations for SITD and PCMCI+ (in seconds).

## A.10 ADDITIONAL EXPERIMENTS

### A.10.1 SYNTHETIC EXPERIMENTS

**Dataset generation.** We first set the number of potential causes $m$, and we fix the lag $\Delta$ and the number of time steps for the dataset $T$. We create a covariate adjacency 3-dimensional tensor $\Sigma$ of dimensions $\Delta \times m \times m$. This tensor has 0-1 coefficients, where $\Sigma_{k,i,j} = 1$ if $X_{t-k}^i$ has a casual effect on $X_t^j$ for all time steps $t$. Similarly, we create an adjacency matrix $\Sigma^{Y}$ for the outcome $Y$. $\Sigma^{Y}$ is a binary $\Delta \times m$ array, such that $\Sigma_{k,j}^{Y} = 1$ if $X_{t-k}^j$ has a causal effect on $Y_t$ for all time steps $t$. The entries of $\Sigma$ and $\Sigma^{Y}$ follow the Bernouli distribution with parameter $p = 0.5$. Note that the resulting casual structure fulfills Axiom (B)-Axiom (D).

We then create $m$ transformations that are used to produce the potential causes $\boldsymbol{X}^1, \ldots, \boldsymbol{X}^m$. Each one of these transformations is modeled by an MLP with 1 hidden layer and 200 hidden units. We use TANH nonlinearities (included also in the output layer) in order to control the scale of the values. The final output value is further scaled up so that all transforms generate values in the range $[-10, 10]$. The input layer of each MLP is coming from the corresponding causal parents of the corresponding time series, as calculated from $\Sigma$.

In order to generate the potential causes $\boldsymbol{X}^i$, we use $\Sigma$ and the MLP transforms. The first value of each time series is randomly generated from a uniform distribution in $[-10, 10]$. Then, each $\boldsymbol{X}^i$ is produced by applying the appropriate transform to its causal parents, as determined by $\Sigma$, and a zero-mean unit variance Gaussian noise is added. We generate the target time series in a similar fashion. Each variable $Y_t$ is produced by applying the MLP transform to its causal parents, as determined by the target adjacency matrix $\Sigma^{Y}$. We then add zero-mean Gaussian noise. The scale of the posterior additive noise for the outcome is referred to as the *noise-to-signal ratio* (NTS).

**Description of the experiments.** We are given a dataset as described above with $m$ potential causes and a fixed NTS for the generation of the outcome $Y$. We determine which series $\boldsymbol{X}^1, \ldots, \boldsymbol{X}^n$ are the causal parents of the outcome, using Algorithm 1. For a given choice of $m$ and NTS, we repeat the experiments five times, and we report on the sample mean for the Accuracy (see Table 3). We additionally report on the F1 Score and CSI Score in Table 5-4 in the Appendix. This experiment is repeated for an increasing number of potential causes, and increasing noise-to-signal ratio, to evaluate the performance of Algorithm 1 on challenging datasets.

In this set of experiments, we learn $\eta_j^i$ as in Line 3 of Algorithm 1, using for the regression task an MLP model. We found that this simple approach, combined with zero-masking, dramatically reduces the false positives of the Student's t-test in Line 13 of Algorithm 1.

**Results.** Table 3 shows the performance of our method with respect to the accuracy, for an increasing number of potential causes and increasing NSR. We observe that our method maintains a good score. These results are partially confirmed when we look at the F1 and CSI scores (see Table 5-4 in the Appendix). In fact, we see that the SITD is stable for an increasingly higher posterior noise.

In Figure 2 we plot the AUROC performance of SITD against RHINO on the synthetic dataset, confirming the competitive performance of STID. In order to calculate the AUROC for SITD, we sort our predictions (for existence of an edge) on the standard deviation of $Z := Y_T \cdot \tilde{g}_j^i + \tilde{\alpha}_j^i \cdot (Y_T - \tilde{g}_j^i) - Y_T \cdot g_j^0 - \alpha_j^0 \cdot (Y_T - g_j^0)$, which is simply the difference of the doubly robust statistics for $\theta^i$ and $\theta^0$ for a datapoint in partition $D_j$.

We provide the CSI Score in Table 4 and the F1 Score in Table 5 for the experiments on synthetic datasets. Recall that in this set of experiments we considered synthetic datasets with varying numbers of potential causes ($m$) and noise-to-signal ratio (NSR). Increasing $m$ and NSR give more challenging settings. We observe that our method maintains good CSI and F1 scores, for increasing NSR.

### A.10.2 PERFORMANCE IN LOW-SAMPLE REGIMES

Here we report the results for additional experiments in the practically important low sample setting. In Figure 3, the plots are provided to support Section 6.2. The double robustness property enables our algorithm to rely on simple estimators with low statistical complexity. As a result, our method shows more consistent performance in low-sample regimes as opposed to existing approaches that are based on overparameterized models demanding so many data points.

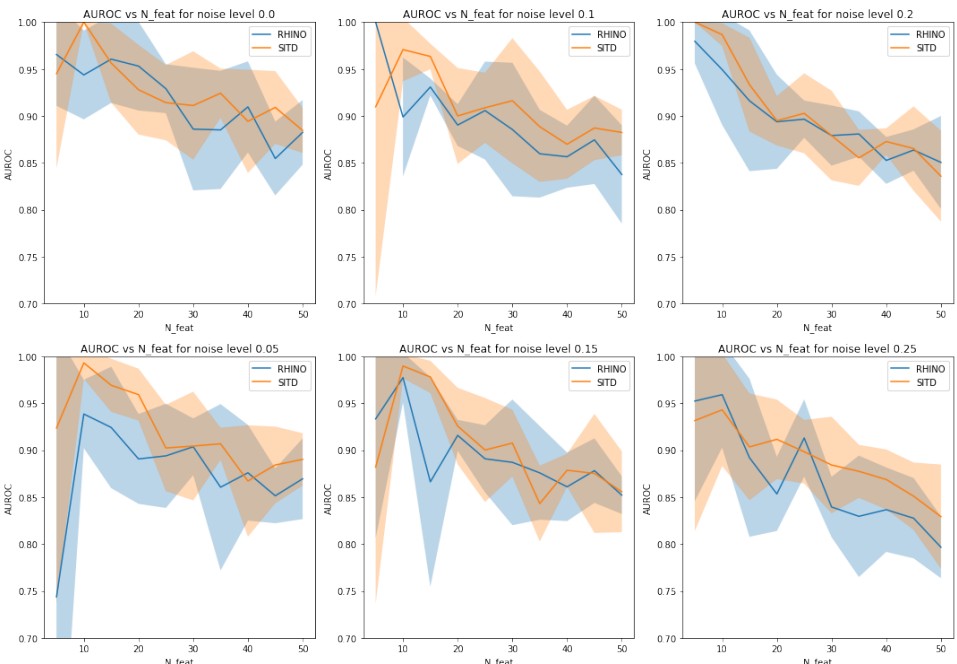

Figure 2: AUROC metric for RHINO and SITD for various noise levels on the synthetic dataset.

Table 3: Accuracy of our method for increasing number of potential causes $m$, and different noise-to-signal ration (NSR). We observe that our method maintains good accuracy, even in challenging settings with many potential causes and high noise.

| | **Accuracy** | | | | | | |
| | **NSR = 0** | **NSR = 0.05** | **NSR = 0.1** | **NSR = 0.15** | **NSR = 0.2** | **NSR = 0.25** | **NSR = 0.3** |
|---|---|---|---|---|---|---|---|
| $m = 5$ | $0.60 \pm 0.09$ | $0.74 \pm 0.25$ | $0.66 \pm 0.19$ | $0.90 \pm 0.11$ | $0.80 \pm 0.06$ | $0.82 \pm 0.10$ | $0.94 \pm 0.12$ |
| $m = 10$ | $0.99 \pm 0.02$ | $0.92 \pm 0.08$ | $0.97 \pm 0.04$ | $0.79 \pm 0.18$ | $0.89 \pm 0.06$ | $0.94 \pm 0.04$ | $0.90 \pm 0.08$ |
| $m = 15$ | $0.89 \pm 0.05$ | $0.89 \pm 0.05$ | $0.88 \pm 0.10$ | $0.85 \pm 0.02$ | $0.79 \pm 0.11$ | $0.79 \pm 0.09$ | $0.81 \pm 0.03$ |
| $m = 20$ | $0.83 \pm 0.07$ | $0.73 \pm 0.05$ | $0.72 \pm 0.07$ | $0.77 \pm 0.10$ | $0.73 \pm 0.05$ | $0.75 \pm 0.05$ | $0.69 \pm 0.06$ |
| $m = 25$ | $0.76 \pm 0.04$ | $0.72 \pm 0.10$ | $0.71 \pm 0.06$ | $0.63 \pm 0.08$ | $0.68 \pm 0.04$ | $0.71 \pm 0.07$ | $0.66 \pm 0.04$ |
| $m = 30$ | $0.76 \pm 0.04$ | $0.74 \pm 0.04$ | $0.72 \pm 0.07$ | $0.70 \pm 0.10$ | $0.66 \pm 0.05$ | $0.68 \pm 0.04$ | $0.64 \pm 0.07$ |
| $m = 35$ | $0.68 \pm 0.02$ | $0.65 \pm 0.07$ | $0.72 \pm 0.06$ | $0.65 \pm 0.03$ | $0.66 \pm 0.05$ | $0.61 \pm 0.07$ | $0.64 \pm 0.09$ |
| $m = 40$ | $0.64 \pm 0.03$ | $0.69 \pm 0.05$ | $0.67 \pm 0.05$ | $0.62 \pm 0.03$ | $0.63 \pm 0.06$ | $0.58 \pm 0.07$ | $0.60 \pm 0.07$ |
| $m = 45$ | $0.65 \pm 0.04$ | $0.68 \pm 0.08$ | $0.58 \pm 0.04$ | $0.61 \pm 0.03$ | $0.64 \pm 0.04$ | $0.59 \pm 0.04$ | $0.61 \pm 0.06$ |
| $m = 50$ | $0.68 \pm 0.05$ | $0.63 \pm 0.05$ | $0.64 \pm 0.06$ | $0.63 \pm 0.05$ | $0.66 \pm 0.05$ | $0.59 \pm 0.08$ | $0.64 \pm 0.07$ |

### A.10.3 RUNTIME AND HARDWARE

To give a taste of the computational efficiency of our method, here we report the average runtime of SITD and the competitive rival Rhino for experiment Table 1 across all five tasks (E.Coli 1, E.Coli 2, Yeast 1, Yeast 2 and Yeast 3) in Table 6. Despite having access only to a single CPU in contrast to the GPU-equipped execution of Rhino our method is almost 20x faster. This is because of the fact that SITD algorithm will provide reasonable results even when employing simple fast efficient estimators (in this case a kernel regression).

Table 4: CSI Score of Algorithm 1 for increasing number of potential causes $m$, and different noise-to-signal ration (NSR). Again, we observe that our method is robust to increasing NSR.

| | CSI Score | | | | | | |
|---|---|---|---|---|---|---|---|
| | NSR = 0 | NSR = 0.05 | NSR = 0.1 | NSR = 0.15 | NSR = 0.2 | NSR = 0.25 | NSR = 0.3 |
| $m = 5$ | $0.57 \pm 0.07$ | $0.71 \pm 0.28$ | $0.63 \pm 0.19$ | $0.86 \pm 0.12$ | $0.69 \pm 0.10$ | $0.73 \pm 0.12$ | $0.91 \pm 0.17$ |
| $m = 10$ | $0.98 \pm 0.04$ | $0.86 \pm 0.14$ | $0.95 \pm 0.06$ | $0.69 \pm 0.21$ | $0.80 \pm 0.09$ | $0.87 \pm 0.07$ | $0.82 \pm 0.14$ |
| $m = 15$ | $0.80 \pm 0.07$ | $0.77 \pm 0.11$ | $0.75 \pm 0.16$ | $0.68 \pm 0.06$ | $0.59 \pm 0.19$ | $0.57 \pm 0.15$ | $0.59 \pm 0.03$ |
| $m = 20$ | $0.66 \pm 0.12$ | $0.52 \pm 0.12$ | $0.46 \pm 0.07$ | $0.56 \pm 0.10$ | $0.46 \pm 0.08$ | $0.51 \pm 0.07$ | $0.39 \pm 0.09$ |
| $m = 25$ | $0.51 \pm 0.06$ | $0.45 \pm 0.12$ | $0.43 \pm 0.09$ | $0.37 \pm 0.09$ | $0.38 \pm 0.05$ | $0.41 \pm 0.08$ | $0.33 \pm 0.06$ |
| $m = 30$ | $0.47 \pm 0.05$ | $0.50 \pm 0.07$ | $0.46 \pm 0.12$ | $0.38 \pm 0.08$ | $0.35 \pm 0.08$ | $0.34 \pm 0.05$ | $0.31 \pm 0.07$ |
| $m = 35$ | $0.42 \pm 0.04$ | $0.31 \pm 0.06$ | $0.39 \pm 0.07$ | $0.29 \pm 0.07$ | $0.30 \pm 0.09$ | $0.22 \pm 0.09$ | $0.26 \pm 0.09$ |
| $m = 40$ | $0.32 \pm 0.06$ | $0.38 \pm 0.05$ | $0.33 \pm 0.08$ | $0.29 \pm 0.06$ | $0.24 \pm 0.06$ | $0.20 \pm 0.07$ | $0.19 \pm 0.11$ |
| $m = 45$ | $0.33 \pm 0.10$ | $0.34 \pm 0.07$ | $0.20 \pm 0.02$ | $0.22 \pm 0.05$ | $0.25 \pm 0.03$ | $0.20 \pm 0.05$ | $0.19 \pm 0.06$ |
| $m = 50$ | $0.33 \pm 0.03$ | $0.29 \pm 0.06$ | $0.29 \pm 0.06$ | $0.23 \pm 0.04$ | $0.26 \pm 0.06$ | $0.21 \pm 0.08$ | $0.26 \pm 0.07$ |

Table 5: F1 Score of the SITD for increasing number of potential causes $m$, and different noise-to-signal ration (NSR). Interestingly, our method maintains a good F1 score for increasing NSR.

| | F1 Score | | | | | | |
|---|---|---|---|---|---|---|---|
| | NSR = 0 | NSR = 0.05 | NSR = 0.1 | NSR = 0.15 | NSR = 0.2 | NSR = 0.25 | NSR = 0.3 |
| $m = 5$ | $0.73 \pm 0.06$ | $0.79 \pm 0.20$ | $0.75 \pm 0.12$ | $0.92 \pm 0.07$ | $0.81 \pm 0.07$ | $0.84 \pm 0.08$ | $0.95 \pm 0.11$ |
| $m = 10$ | $0.99 \pm 0.02$ | $0.92 \pm 0.09$ | $0.98 \pm 0.03$ | $0.80 \pm 0.15$ | $0.89 \pm 0.06$ | $0.93 \pm 0.04$ | $0.90 \pm 0.09$ |
| $m = 15$ | $0.89 \pm 0.05$ | $0.86 \pm 0.08$ | $0.85 \pm 0.11$ | $0.81 \pm 0.04$ | $0.73 \pm 0.15$ | $0.72 \pm 0.12$ | $0.74 \pm 0.03$ |
| $m = 20$ | $0.79 \pm 0.08$ | $0.67 \pm 0.10$ | $0.62 \pm 0.07$ | $0.71 \pm 0.09$ | $0.63 \pm 0.07$ | $0.67 \pm 0.06$ | $0.56 \pm 0.09$ |
| $m = 25$ | $0.68 \pm 0.05$ | $0.61 \pm 0.11$ | $0.60 \pm 0.09$ | $0.53 \pm 0.11$ | $0.55 \pm 0.05$ | $0.57 \pm 0.08$ | $0.49 \pm 0.07$ |
| $m = 30$ | $0.64 \pm 0.05$ | $0.66 \pm 0.06$ | $0.62 \pm 0.10$ | $0.54 \pm 0.09$ | $0.51 \pm 0.09$ | $0.50 \pm 0.06$ | $0.46 \pm 0.09$ |
| $m = 35$ | $0.59 \pm 0.05$ | $0.47 \pm 0.07$ | $0.56 \pm 0.07$ | $0.45 \pm 0.08$ | $0.46 \pm 0.11$ | $0.35 \pm 0.11$ | $0.41 \pm 0.11$ |
| $m = 40$ | $0.49 \pm 0.07$ | $0.55 \pm 0.05$ | $0.49 \pm 0.08$ | $0.45 \pm 0.07$ | $0.38 \pm 0.08$ | $0.33 \pm 0.09$ | $0.30 \pm 0.15$ |
| $m = 45$ | $0.49 \pm 0.11$ | $0.50 \pm 0.08$ | $0.34 \pm 0.03$ | $0.36 \pm 0.06$ | $0.40 \pm 0.04$ | $0.33 \pm 0.06$ | $0.32 \pm 0.08$ |
| $m = 50$ | $0.50 \pm 0.03$ | $0.44 \pm 0.07$ | $0.44 \pm 0.08$ | $0.38 \pm 0.05$ | $0.41 \pm 0.07$ | $0.34 \pm 0.11$ | $0.40 \pm 0.09$ |

Table 6: Runtime and Hardware used for our method (SITD) and the state-of-the-art baseline Rhino.

| Category\Method | Rhino | SITD (ours) |
|---|---|---|
| Runtime | 18 mins 40 sec $\pm$ 30 sec | 57.12 sec $\pm$ 1.6 sec |
| Hardware | 1 NVIDIA A100 GPU + AMD EPYC 7402 24-Core CPU | 11th Gen Core i5-1140F CPU |

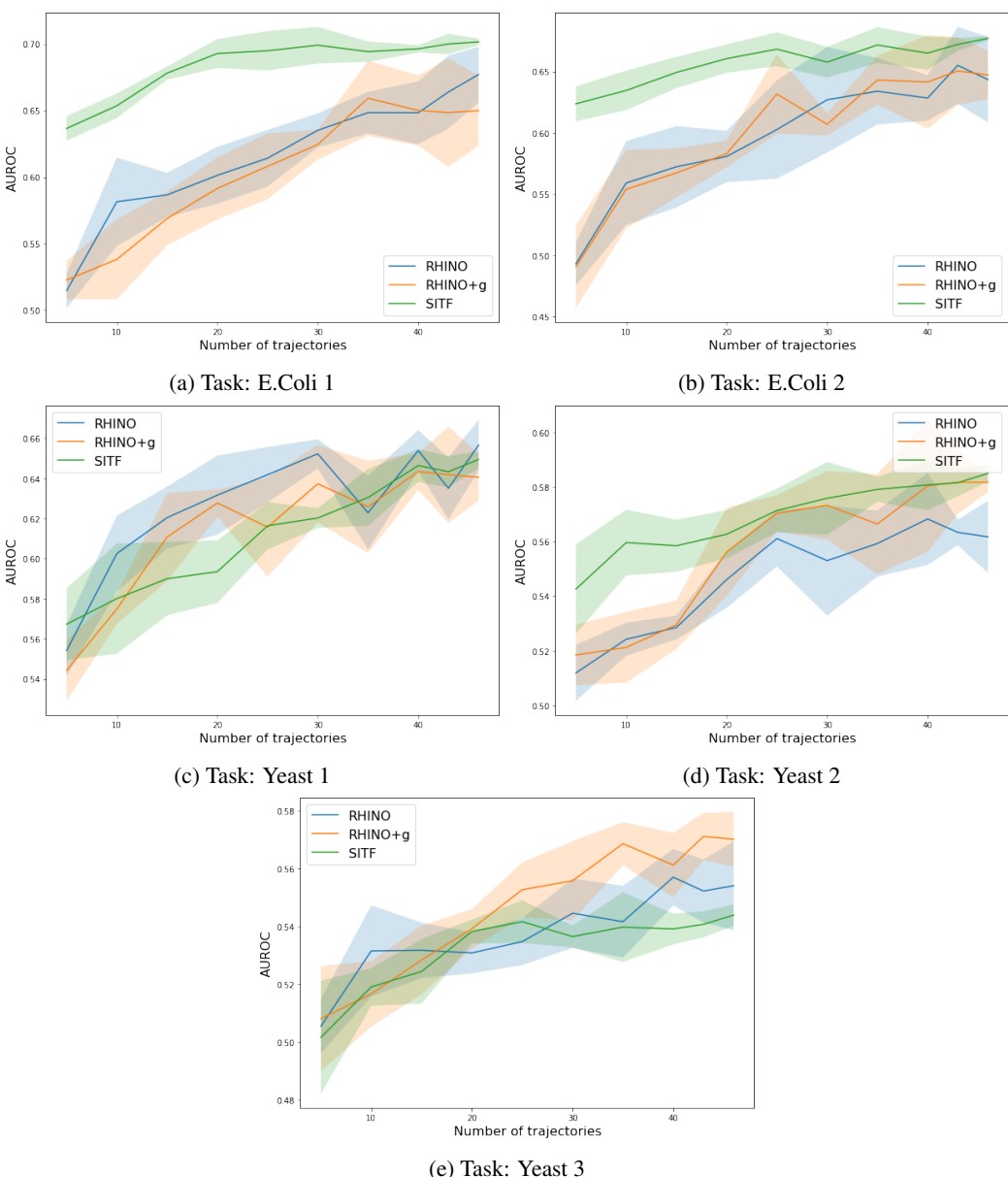

(a) Task: E.Coli 1

(b) Task: E.Coli 2

(c) Task: Yeast 1

(d) Task: Yeast 2

(e) Task: Yeast 3

Figure 3: This figure demonstrates the consistent performance of SITD w.r.t number of observations compared to state-of-the-art methods Rhino and Rhino+g. Note that Rhino and Rhino+g are built on neural networks. SITD significantly outperforms Rhino and Rhino+g in E.Coli 1 and E.Coli 2 and shows competitive results in Yeast 1. Thanks to the double robustness property of SITD, the dependence of our algorithm on the estimator is much lower than the well-established approaches. In this regard, SITD with a simple kernel regression with polynomial kernels has superior performance compared to state-of-the-art methods Rhino and Rhino+g. This superiority gets magnified in the low number of observation regimes due to the high sample complexity required by Rhino and Rhino+g.

