# OpenReview forum: "Doubly Robust Structure Identification from Temporal Data"
_ICLR.cc/2024/Conference — Submitted to ICLR 2024_

### Official Review · Reviewer_biK3 · 2023-10-30

**Soundness:** 3 good
**Presentation:** 2 fair
**Contribution:** 3 good
**Rating:** 3
**Confidence:** 3

**Summary:**

The paper proposes an approach for learning time varying causal features of a target variable using Granger causality and doubly robust methods. The approach can also be used for full causal discovery and does not require the faithfulness or causal sufficiency assumptions.

A theorem is given that under the assumptions made, Granger causality is equivalent to true causation. The approach then proceeds by choosing a target causal feature, fitting parameters and testing significance of the parameters.

The approach is empirically evaluated on a semisynthetic dataset (Dream3) and is one of the top performing methods in 3/5 of the experiments.

**Strengths:**

1) The approach is very novel to the best of my knowledge

2) If the theorem that under the given setting, Granger causality is equivalent to true causation is correct (I am unable to check the proof of this theorem in the appendix), then the approach is sound

3) Background and related work are extensively reviewed

**Weaknesses:**

1) Generally speaking, the paper is hard to follow and the goals of the proposed method are unclear given the entire paper. The approach is motivated as to be for (time-varying) causal feature selection for a target variable. However, it is claimed in the paper that it can also be used for full causal discovery, but it's not clear the evaluation is for either causal feature selection or full causal discovery.

2) The task performed in the evaluation section is not described at all. Presumably the task should be causal feature selection, but all the reader is told is the metric used for evaluation is AUC, which doesn't sound like we're evaluating the correct set of causal features. Furthermore, the methods used in the evaluation section do not appear to be causal feature selection methods and are different from the related methods mentioned earlier in the paper.

3) Aside from the above confusion about the evaluation section, the empirical work is minimal in general and standard errors are not included.

**Questions:**

1) Can the authors explain the evaluation task? Is it causal feature selection? What is the actual target that AUC is reported for?

2) How is the time-varying nature accounted for in the evaluation?

3) Why are the baselines in the evaluation section different from the methods mentioned in the introduction for the problem the method is proposed for? Has the approach been compared to the other causal feature selection methods mentioned earlier in the paper?

4) Why are standard errors missing form the baselines? Is the improvement significant?

5) Is there a limitation when extending the approach to the full causal discovery setting?

---

> ### Author Response · Authors · 2023-11-17
> **Clarification**
>
> We appreciate the reviewer for evaluating our paper and acknowledging its novelty. We address each of your concerns in the following and look forward to further discussion if you have additional feedback. We are committed to making the necessary revisions to enhance the clarity, accessibility, and overall coherence of our writing.
>
>
> > Generally speaking, the paper is hard to follow and the goals of the proposed method are unclear given the entire paper. The approach is motivated as to be for (time-varying) causal feature selection for a target variable. However, it is claimed in the paper that it can also be used for full causal discovery, but it's not clear the evaluation is for either causal feature selection or full causal discovery.
>
> Thank you for your comment! If you have specific suggestions on how to improve the write-up, we will gladly incorporate your suggestions in our manuscript.
>
> To answer your question, in the experiments, we consider both causal feature selection and causal discovery. The reason why we consider both scenarios is to ensure a fair comparison with competitors. Note that causal feature selection for timeseries data is relatively underexplored. Hence, we compared our method with causal structure learning algorithms to showcase performance of our method with stronger evidence. In Section 6.1 we focus on causal discovery, whereas in Section 6.2 / Appendix J.1 we focus on causal feature selection. We are happy to make this point clearer in our submission.
>
> > The task performed in the evaluation section is not described at all. Presumably the task should be causal feature selection, but all the reader is told is the metric used for evaluation is AUC, which doesn't sound like we're evaluating the correct set of causal features. Furthermore, the methods used in the evaluation section do not appear to be causal feature selection methods and are different from the related methods mentioned earlier in the paper.
>
>
>
> Yes, thank you for pointing it out! Our experimental framework essentially replicates previous related work. In our experimental section, we report on all state-of-the-art benchmarks for the DREAM3 datasets. Furthermore, the use of the AUC is consistent with previous related work [1] (state-of-the-art). There are several reasons for the usage of AUC other than other metrics like accuracy, F1 score, recall, etc, when comparing the performance of statistical algorithms that involve statistical tests. All of those metrics need a p-value thresholding which may lead to the following potential problems in the comparisons: vulnerability to class imbalance, task-specific threshold choices, model sensitivity to false positives and negatives, sensitivity to class probabilities, etc.
>
>
>
> > Aside from the above confusion about the evaluation section, the empirical work is minimal in general and standard errors are not included.
>
> Yes, thank you for pointing it out. Our experimental framework considers a variety of baselines, and it is consistent with previous related work [1,2,3,4]. Throughout our experimental investigaion, we did not find that standard errors play a significant role, as it was also observed in related work [1]. Please inform us of other experiments/datasets/plots that you would think that our manuscript can benefit from, so we can incorporate your comments in our experiment section.
>
> > Can the authors explain the evaluation task? Is it causal feature selection? What is the actual target that AUC is reported for?
>
> In the semi-synthetic experiment involving the Dream3 dataset (section 6.1, Appendix J.2), our objective is full causal discovery. Each of the five tasks within Dream3 involves observing the evolution of a system comprising 100 genes over time, subjected to 46 different perturbations. The ground truth causal graph delineating the relationships among these genes is provided. Out of the $100*99$ possible directed edges among the genes, some are indeed present, indicating direct causal influence, while others are not. This forms the actual target for AUROC calculations.
>
> In the additional synthetic experiments (Appendix J.1), the focus shifts to causal feature selection. Here, the ground truth used in the AUROC metric is represented by the $\Delta \times m$ binary adjacency matrix $\Sigma^{\mathbf{Y}}$, where $\Delta$ signifies the lag, and $m$ represents the number of potential causes."

---

> > ### Author Response · Authors · 2023-11-17
> > **Clarification - Continued**
> >
> > >How is the time-varying nature accounted for in the evaluation?
> >
> > In our experiments, we adopt the stationarity assumption alongside a fixed lag. To account for the time-varying nature of the dataset, we carefully select a suitable lag. It's important to highlight that while the stationarity assumption is included in the literature for experimental convenience [1,2,3,4], it is not a necessary condition for the application of our method.
> >
> > >Why are the baselines in the evaluation section different from the methods mentioned in the introduction for the problem the method is proposed for? Has the approach been compared to the other causal feature selection methods mentioned earlier in the paper?
> >
> > In the introduction, we give an overview of the Causal Feature Selection problem, by discussing many previous related algorithms that have been propsed in recent years. Moreover, we broadly discuss previous related work. In the experimental section, we focus on the main competitors, particularly on the DREAM3 datasets. We remark that our experiments are consistent with previous related work [1,2,3,4].
> >
> > >Why are standard errors missing form the baselines? Is the improvement significant?
> >
> > The baseline results in Table 1 are reported directly from [1,2], where the standard errors are also missing.
> >
> > As noted in [1], RHINO is very stable and the standard error is minimal. This is also true for our propsed method. Hence, the improvement in our experimental framework can be considered significant. Please take into account the computational aspects that our method using simple regressions outperforms deep learning-based approaches that necessitate substantial computational resources, particularly GPUs.
> >
> >
> > >Is there a limitation when extending the approach to the full causal discovery setting?
> >
> > Our method naturally extends to full causal discovery as it is. In order for the proposed method to work, Axioms $(A)$ to $(C)$ must hold for any choice of the target variable Y. In the full causal discovery setting, this implies that there are no confounding effects or instantaneous effects between any of the time series of the model.
> >
> > [1] Gong, Wenbo, Joel Jennings, Cheng Zhang, and Nick Pawlowski. “Rhino: Deep Causal Temporal Relationship Learning with History-dependent Noise.” In The Eleventh International Conference on Learning Representations. 2022.
> >
> > [2] Saurabh Khanna and Vincent YF Tan. Economy statistical recurrent units for inferring nonlinear granger causality. arXiv preprint arXiv:1911.09879, 2019.
> >
> > [3] Roxana Pamfil, Nisara Sriwattanaworachai, Shaan Desai, Philip Pilgerstorfer, Konstantinos Geor- gatzis, Paul Beaumont, and Bryon Aragam. Dynotears: Structure learning from time-series data. In International Conference on Artificial Intelligence and Statistics, pp. 1595–1605. PMLR, 2020.
> >
> > [4] Alex Tank, Ian Covert, Nicholas Foti, Ali Shojaie, and Emily Fox. Neural granger causality for nonlinear time series. stat, 1050:16, 2018.

---

> > > ### Author Response · Authors · 2023-11-20
> > > **Anymore Concerns?**
> > >
> > > Dear Reviewer,
> > >
> > > We sincerely appreciate the time you dedicated to evaluating our work. We would like to inquire if our rebuttal adequately addressed your concerns and if we can provide any additional information that you think is necessary.
> > >
> > > Thank you for your consideration!

---

### Official Review · Reviewer_fyvt · 2023-10-31

**Soundness:** 2 fair
**Presentation:** 2 fair
**Contribution:** 2 fair
**Rating:** 3
**Confidence:** 3

**Summary:**

The paper proposes an algorithm to discover causal relationship using time series data. It is based on the double/debiased machine learning (DML) framework that has been popular in the recent literature. There are two main theoretical results: (I) Theorem 4.1 shows that under a set of axioms (A to C, in particular), true causality is equivalent to Granger causality, and (ii) Theorem 4.2 claims that under axioms A to D, Granger causality is equivalent to checking whether two expectations are the same or not. The algorithm called Structure Identification from Temporal Data (SITD) is given on page 7 and its numerical performance is illustrated using the Dream3 dataset.

**Strengths:**

The research question addressed in the paper is of very high importance. As mentioned in the first paragraph on page 1, there are numerous scientific fields where causality questions need be addressed with time series data.

**Weaknesses:**

1. The paper focuses on time series data but there is no statistical analysis focusing on time series data. For example, on page 7, it is stated that "Under mild conditions on the convergence of $g_j^0$, $g_j^i$ and $\alpha_j^0$, $\alpha_j^i$, the quantity $\theta^0 − \theta^i$ has $\sqrt{n}$-consistency" and that "We refer the reader to Chernozhukov et al. (2022; 2018) for a proof of the $\sqrt{n}$-consistency for estimates as $\theta^0$ and $\theta^i$."  I do not think the cited references deal with time series data directly. It is disappointing that the paper does not provide any extensive treatment of time series analysis.

2. Lemma A.1 claims that conditional mean independence in part 1 is equivalent to the conditional dependence in part 2. This seems mainly driven by Axiom (A) where the error $\varepsilon$ is exogenous independent noise. I feel that this is a rather restricted setting. For example, suppose that Y is the time series of financial returns (e.g., S&P 500) and X is the causal factor that does not affect the conditional mean of returns but does affect the conditional variance of returns (typically called volatility in finance). It seems that the framework in the current paper excludes this kind of scenario. It is unclear to me what sense Axiom (A) is necessary; related to this point, Appendix A.3 is difficult to understand (see question 1 below).

**Questions:**

1. Appendix A.3 is difficult to understand. What are roles of $W_t$ and $Z_t$? $\Sigma$ is not a positive definite matrix here and seems too irregular. Some further comments would be useful.

2. The derivation on page 18 after "We now prove the claim" is difficult to follow. It seems to me that it is already assumed that $E[Y_ T | X_t^i = x, I_T^{\backslash i} = i] = E[Y_T | X_t^i =x', I_T^{\backslash i} = i]$ for any $x$ and $x'$ in the derivation; but I am not sure why. Does the current proof imply the if and only if result for equation (3)? Some clarifications would be helpful.

3. I cannot follow why equation (4) is a good property. This indicates that the bias multiplied by $\sqrt{n}$ goes to zero. It might be better to show that the root mean squared error multiplied by $\sqrt{n}$ goes to zero as $n \rightarrow \infty$. Some explanations would be helpful.

4. In the experiments on page 8, the area under the ROC curve (AUROC) is used as the performance metric. It would be beneficial why this metric is related to causality concerns.

[Update after the discussion period] The author(s) provided timely responses to my comments/questions; I very much appreciate them; however, I still have concerns and would like to keep my ratings.

---

> ### Author Response · Authors · 2023-11-17
> **Clarification**
>
> We express our gratitude for reviewer's interest and constructive feedback regarding the paper’s clarity. We address your concerns in the following. We look forward to your further feedback to make sure the contributions of the paper are clarified clearly.
>
> > The paper focuses on time series data but there is no statistical analysis focusing on time series data. For example, on page 7, it is stated that "Under mild conditions on the convergence of $g^0_j$, $g^i_j$, and $\alpha_j^0, \alpha_j^i$, the quantity $\theta^0 - \theta^i$ has $\sqrt{{n}}$-consistency" and that "We refer the reader to Chernozhukov et al. (2022; 2018) for a proof of the
> $\sqrt{{n}}$-consistency for estimates as $\theta^0$ and $\theta^i$." I do not think the cited references deal with time series data directly. It is disappointing that the paper does not provide any extensive treatment of time series analysis.
>
> Please note that, even though the data has time series structure, but i.i.d. trajectories of the data are provided and each trajectory can be treated as a single draw from an i.i.d. distribution. Hence, the strong consistency of our method follows directly from Chernozhukov et al. (2022; 2018). In our paper, we only provide an informal discussion on this point, to avoid a complicated notation that would have made the paper difficult to read. However, we are happy to provide a formal statment and an in-depth discussion of this point, if required. It is important to note that double ML can be extended to handle data of stochastic nature (Markovian non-i.i.d.) by applying it iteratively in a backward order, starting from the end of the process and moving towards the beginning. At each time step, following the doubly robust estimation, residuals of causal effects are computed to update the auxiliary random variables from the preceding time step. It can be demonstrated that the resulting scores derived from these auxiliary random variables satisfy the Neyman Orthogonality condition. This way the Markovian structure of the stochastic process at each step is cancelled out. The algorithm then proceeds to the previous time step. For further details, please refer to [1]. A similar approach can be employed here with appropriate adjustments, but delving into those modifications is beyond the scope of the main message conveyed in this manuscript.
>
> > Lemma A.1 claims that conditional mean independence in part 1 is equivalent to the conditional dependence in part 2. This seems mainly driven by Axiom (A) where the error
>  is exogenous independent noise. I feel that this is a rather restricted setting. For example, suppose that Y is the time series of financial returns (e.g., S&P 500) and X is the causal factor that does not affect the conditional mean of returns but does affect the conditional variance of returns (typically called volatility in finance). It seems that the framework in the current paper excludes this kind of scenario. It is unclear to me what sense Axiom (A) is necessary; related to this point, Appendix A.3 is difficult to understand (see question 1 below).
>
> Axiom (A) essentially requires that no confounding effect exists between the target variable Y and the causal parents. This assuption is essential for identifiability. As you correctly observe, this setting is restrictive. However, without any assumption we simply cannot get identifiability. Conditional mean independence is a necessary assumption in this setting. In other words, if this assumption fails, as in the example that you provide, than no method will provably be able to recover the true underlying causal structure due to identifiability issues. Please refer to appendix A.3 for an example of this identifiability problem without Axiom (A).
>
> >Appendix A.3 is difficult to understand. What are roles of $W_t$ and $Z_t$? $\Sigma$ is not a positive definite matrix here and seems too irregular. Some further comments would be useful.
>
> $W_t$ and $Z_t$ are simply two time series, much like $X_t$ and $Y_t$. We use different notation, to highlight that the time series are generated differently. The matrix $\Sigma$ is a 2x2 positive semi-definite matrix, with eigenvalues lambda=2 and lambda=0. We hope this clarifies the point?

---

> > ### Author Response · Authors · 2023-11-17
> > **Clarification - Continued**
> >
> > >The derivation on page 18 after "We now prove the claim" is difficult to follow. It seems to me that it is already assumed that $E[Y_ T | X_t^i = x, I_T^{\backslash i} = i] = E[Y_T | X_t^i =x', I_T^{\backslash i} = i]$ for any $x$ and $x'$ in the derivation; but I am not sure why. Does the current proof imply the if and only if result for equation (3)? Some clarifications would be helpful.
> >
> > On Page 18, our goal is not to prove that $E[Y_ T | X_t^i = x, I_T^{\backslash i} = i] = E[Y_T | X_t^i =x', I_T^{\backslash i} = i]$. Rather, we prove that the statistical test can be obtained as the difference of two linear moment functionals. Essentially, the contribution of that proof is the last chain of inequaities. We are happy to make this point clearer in our submission.
> >
> > >I cannot follow why equation (4) is a good property. This indicates that the bias multiplied by $\sqrt{n}$ goes to zero. It might be better to show that the root mean squared error multiplied by $\sqrt{n}$ goes to zero as $n \rightarrow \infty$. Some explanations would be helpful.
> >
> > Equation (4) is a standard strong convergence property, which follows from our analysis. This is indeed a good property, in the sense that the error between the true term and our approximation "quickly" converges to zero, for an increasing number of samples. Another way to express strong consistency is that as the sample size $n$ approaches infinity, $\sqrt{n}(\theta^0 - \theta^j)$ converges in distribution to $\mathcal{N}(0, \sigma^2)$, where the variance term $\sigma^2$ is independent of the sample size $n$.
> >
> > >In the experiments on page 8, the area under the ROC curve (AUROC) is used as the performance metric. It would be beneficial why this metric is related to causality concerns.
> >
> > Regarding the experiments with Dream3, we mention that it is the standard practice to measure the performance of a new method on this benchmark using the AUROC metric.
> >
> > This metric is applicable to binary classification tasks; all the predictions are sorted from most likely to be positive to least likely, and the AUROC measures the ability of the model to correctly rank these predictions.
> >
> > In the Dream3 challenge, the task is gene network inference, so researchers assess how well their methods can identify true causal relationships amidst many potential but non-existent ones. Given the inherent binary classification nature of causal inference (an edge of the gene network either exists or not), the AUROC metric is more appropriate than other metrics (e.g. accuracy, F1 score) to distinguish between the two classes for this particular dataset. More specifically, Dream 3 is characterized by class imbalance; gene networks are very sparse. For example in the E. Coli 1 task of Dream 3, out of  the 100*99 potential directed edges between the 100 genes, less than 200 were present. AUROC is less sensitive to class imbalance compared to e.g accuracy, because the we simply sort the predictions about existence/absence of an edge according to how much confident our causal method is; that is, only the relative order matters. Furthermore, in this particular domain, a.k.a gene network inference, it's often more critical to correctly rank pairs of genes by their likelihood of having a causal relationship than to classify each pair definitively. AUROC evaluates a model's ability to rank positive cases higher than negative ones, which is directly aligned with this requirement.
> >
> > [1] Lewis, Greg, and Vasilis Syrgkanis. "Double/Debiased Machine Learning for Dynamic Treatment Effects." In NeurIPS, pp. 22695-22707. 2021.

---

> > > ### Author Response · Authors · 2023-11-20
> > > **Anymore Concerns?**
> > >
> > > Dear Reviewer,
> > >
> > > We sincerely appreciate the time you dedicated to evaluating our work. We would like to inquire if our rebuttal adequately addressed your concerns and if we can provide any additional information that you think is necessary.
> > >
> > > Thank you for your consideration!

---

> > > > ### Comment · Reviewer_fyvt · 2023-11-22
> > > >
> > > > I would like to thank the author(s) for providing detailed responses. However, my concerns still remain at large. First, it would be very demanding to restrict the scope to i.i.d. data in the context of time series/temporal data. Adding Markov or other types of time series assumptions would require a substantial additional analysis. Second, to my reading, equation (4) is a good property only in the sense that the __expected__ error (that is, the __mean__ of the error) converges to zero; however, its variance may not necessarily go to zero.

---

> > > > > ### Author Response · Authors · 2023-11-22
> > > > > **Strong Consistency is very Desirable!**
> > > > >
> > > > > Thank you very much for participating in the discussion!
> > > > >
> > > > > > I would like to thank the author(s) for providing detailed responses. However, my concerns still remain at large. First, it would be very demanding to restrict the scope to i.i.d. data in the context of time series/temporal data. Adding Markov or other types of time series assumptions would require a substantial additional analysis.
> > > > >
> > > > > As mentioned in our previous reply, we only assume that the trajectories of the time series are sampled i.i.d. This assumption, does not pose additional structural requirements on the time series themselves. Consider a stochastic process $\\{X_t\\}_{t=1}^T$ with a finite-time horizon T. A trajectory for this stochastic process is just a sequence of samples
> > > > >
> > > > > $x_1, \dots, x_t, \dots, x_{T}$ collected at each time step. Our dataset consists of such sequences, sampled independently and identically (i.i.d.). In our analysis, we do not assume that samples within each sequence are i.i.d. Hence, our framework extends to Markov chains or more complex stochastic processes. In the end, we note that almost all of the existing literature is built on the i.i.d. trajectories assumption [1-6].
> > > > >
> > > > >
> > > > > > Second, to my reading, equation (4) is a good property only in the sense that the expected error (that is, the mean of the error) converges to zero; however, its variance may not necessarily go to zero.
> > > > >
> > > > > As previously explained, strong consistency is represented by $\sqrt{n}(\theta^0 - \theta^j) \overset{\mathrm{d}}{\rightarrow} \mathcal{N}(0, \sigma^2)$ (where $\overset{\mathrm{d}}{\rightarrow}$ denotes convergence in distribution). This expression can be equivalently expressed as $(\theta^0 - \theta^j) \overset{\mathrm{d}}{\rightarrow} \mathcal{N}(0, \frac{\sigma^2}{n})$, indicating that the **variance** of the estimation error tends to zero as the sample size ($n$) approaches infinity. To provide further clarification, the guarantees of the **Central Limit Theorem** share a similar nature, specifically, $\sqrt{n} (\frac{1}{n}\sum_{i = 1}^{n} X_i - \mathbb{E}[X]) \overset{\mathrm{d}}{\rightarrow} \mathcal{N}(0, \sigma^2)$, which is indeed a desirable property.
> > > > >
> > > > > We sincerely appreciate your concerns and your engagement in the discussion. We are happy to further continue the discussion.
> > > > >
> > > > > [1] Gong, Wenbo, Joel Jennings, Cheng Zhang, and Nick Pawlowski. “Rhino: Deep Causal Temporal Relationship Learning with History-dependent Noise.” In The Eleventh International Conference on Learning Representations. 2022.
> > > > >
> > > > > [2] Jakob Runge, Peer Nowack, Marlene Kretschmer, Seth Flaxman, and Dino Sejdinovic. Detecting and quantifying causal associations in large nonlinear time series datasets. Science advances, 5(11):eaau4996, 2019.
> > > > >
> > > > > [3] Jakob Runge. Discovering contemporaneous and lagged causal relations in autocorrelated nonlinear time series datasets. In Conference on Uncertainty in Artificial Intelligence, pages 1388–1397. PMLR, 2020.
> > > > >
> > > > > [4] Andreas Gerhardus and Jakob Runge. High-recall causal discovery for autocorrelated time series with latent confounders. Advances in Neural Information Processing Systems, 33:12615–12625, 2020.
> > > > >
> > > > > [5] Bussmann, Bart, Jannes Nys, and Steven Latré. “Neural additive vector autoregression models for causal discovery in time series.” Discovery Science: 24th International Conference, DS 2021, Halifax, NS, Canada, October 11–13, 2021, Proceedings 24. Springer International Publishing, 2021.
> > > > >
> > > > > [6] Hyvärinen, A., Zhang, K., Shimizu, S., & Hoyer, P. O. (2010). Estimation of a structural vector autoregression model using non-gaussianity. Journal of Machine Learning Research, 11(5).

---

> > > > > > ### Comment · Reviewer_fyvt · 2023-11-22
> > > > > >
> > > > > > Thank you for your quick responses. For the first response, perhaps I might be mistaken regarding the sampling structure. More importantly, my original comment simply points out that the current paper does not have fully established the desired statistical results; it is unclear to me whether they can be obtained easily from Chernozhukov et al. (2022; 2018). For the second response, I agree that if you have asymptotic normality as you described, you will have $\sqrt{n}$-consistency. As a side remark, strong consistency typically refers to the situation where a sequence of estimators converges almost surely to the true value of the parameter. I find that your use of strong consistency seems different from this convention. In short, I can see that there are many positive parts in the paper but it seems that the paper may need further careful revision.

---

### Official Review · Reviewer_95xj · 2023-11-01

**Soundness:** 3 good
**Presentation:** 3 good
**Contribution:** 2 fair
**Rating:** 6
**Confidence:** 2

**Summary:**

The authors present a doubly robust structure identification method for temporal data that can identify the direct causes of the target variable assuming additive noises, even in the presence of cyclic structures and in the absence of faithfulness or causal sufficiency.

**Strengths:**

1. The authors offer a discussion connecting Granger's causality with Pearl's framework, which is thought-provoking.
2. The authors propose an algorithm based on a parameter estimation framework, namely DoubleML, to detect the causes of the target variable.
3. The literature review is comprehensive.
4. The authors conduct extensive experiments on semi-synthetic and synthetic datasets, compared with several baselines.

**Weaknesses:**

1. In the contribution, the authors claim that the proposed algorithm can be used for full causal discovery under some assumptions. The related discussion in section 5.2 is limited without details.
2. In principle, the approach adheres to steps (1) through (4) in section 4.2, yet the practical algorithm has been adjusted to account for the time-consuming nature of "large instances." While the approach outline aligns with the proven theorem, a gap exists between the outlined approach and the modified algorithm. Is it feasible to implement the approach strictly in smaller instances, adhering to steps (1) through (4)? Furthermore, what does "large instances" imply in this context?
3. There is no real-world application provided in the paper.
4. Regarding the baselines, from my understanding, some of them are designed for full causal discovery, encompassing the detection of causes for target variables and beyond. In contrast, the proposed algorithm primarily focuses on feature detection. In the experiment section, are there any specific modifications necessary to ensure a fair comparison?

**Questions:**

1. Can you please provide a brief explanation of the role played by the causal graph in the proposed algorithm? Personally, I am under the impression that the causal graph is unrelated to the proposed method, rendering the faithfulness assumption, cyclic structure, and causal sufficiency irrelevant to the algorithm. Thus, I do not consider the relaxation of this assumption as an advantage of the method, as it falls outside the scope of the algorithm. Please correct me if I missed the point.
2. In the first equation on page 4, what is $N$?
3. In equation 3, what is $n$? Should it be $k$?
4. I felt lost that in equation 3, $g^0_0$ and $g^i_0$ equal to the same conditional expectation as $\alpha^0_0$ and $\alpha_0^i$ in the second point in section 5.1. Are they the same things?
5. As the appendix states, $k$ ranges from 3 to 7. What is the value of $k$ used in each experiment? Is the algorithm output sensitive to the value of $k$?
6. The term "trajectories" means the time series, correct? In Fig.2, what does $N_feat$ represent? Is $N_feat$ indicative of the number of trajectories? Additionally, in Fig.3, all the algorithms exhibit improved performance with an increase in the number of trajectories. Could you provide a brief explanation for this trend? Moreover, why does Fig.3 depict the performance in low-sample regimes, and how are "low-sample regimes" reflected in the Fig.3?
7. Is there a specific reason for using only one baseline algorithm in the experiments presented in the appendix?

---

> ### Author Response · Authors · 2023-11-17
> **Clarification**
>
> We sincerely appreciate the reviewer’s recognition regarding the significance of the contributions made by this research in the challenging domain of causal structure learning for time-series under highly general assumptions.
>
> > In the contribution, the authors claim that the proposed algorithm can be used for full causal discovery under some assumptions. The related discussion in section 5.2 is limited without details.
>
> Thanks a lot for point this out! This simple reduction was not included in the interest of space and it goes as the following:
>
> If the underlying causal structure doesn't have hidden confounders (noted as "fully-observed" in the manuscript) and doesn't contain any cycles (noted as "acyclic" in the manuscript), then we can iteratively select each node of the causal graph as our target variable and apply our causal feature selction algorithm to find its direct causes. These two conditions are necessary for the Axiom (A) to be true when any node of the causal graph could be selected as the target variable. In the end, the whole causal structure will be discovered.
>
> We will make sure to clarify further in the final version of the manuscript.
>
> > In principle, the approach adheres to steps (1) through (4) in section 4.2, yet the practical algorithm has been adjusted to account for the time-consuming nature of "large instances." While the approach outline aligns with the proven theorem, a gap exists between the outlined approach and the modified algorithm. Is it feasible to implement the approach strictly in smaller instances, adhering to steps (1) through (4)? Furthermore, what does "large instances" imply in this context?
>
> Thanks a lot for your precise and detailed question! When we mention "large instances", we are specifically referring to cases where the number of nodes, denoted as $m$, is substantial.  The principled approach necessitates $\mathcal{O}(m)$ regressions, which is linear in the number of nodes. However, by incorporating the zero-masking heuristic, we can optimize the process, reducing the required regressions to just $\mathcal{O}(1)$. This proves particularly advantageous in situations where computational resources are limited, but the causal graph is huge. It's essential to highlight that, as demonstrated in Appendix A.7, there is no discernible difference between the principled approach and the one utilizing the heuristic when the underlying estimator is linear regression.
>
> > There is no real-world application provided in the paper.
>
> Indeed, thank you for highlighting this consideration. In previous related works [2,3,4,5], Dream3 was employed as a gold-standard benchmark which is of systems' biology nature. However, we acknowledge that over-tuning causal discovery algorithms exclusively for a single dataset could present potential issues. As you aptly observe, other datasets might be pertinent in this context. We are open to the inclusion of any additional datasets you may suggest.
>
> > Regarding the baselines, from my understanding, some of them are designed for full causal discovery, encompassing the detection of causes for target variables and beyond. In contrast, the proposed algorithm primarily focuses on feature detection. In the experiment section, are there any specific modifications necessary to ensure a fair comparison?
>
> There are no modifications necessary to ensure fair comparison. The reason is that our algorithm naturally extends to full causal discovery, without any modifications, by simply treating each variable as a target. The only subtle issue that could arise is the following: since a two-sample Student’s t-test is used to determine whether a particular covariate is a causal parent of the target, then a multiple hypothesis testing issue could arise, because we apply this procedure for every pair of variables. However, we do not face such a problem here because, according to the standard practice, the AUROC metric is used for evaluation in the Dream 3 benchmark. For this metric, we simply sort the predictions about existence/absence of an edge; that is, only the relative order matters.

---

> > ### Author Response · Authors · 2023-11-17
> > **Clarification - Continued**
> >
> > >Can you please provide a brief explanation of the role played by the causal graph in the proposed algorithm? Personally, I am under the impression that the causal graph is unrelated to the proposed method, rendering the faithfulness assumption, cyclic structure, and causal sufficiency irrelevant to the algorithm. Thus, I do not consider the relaxation of this assumption as an advantage of the method, as it falls outside the scope of the algorithm. Please correct me if I missed the point.
> >
> > Our algorithm uncovers the underlying causal structure of the DGP. Hence, the causal graph is very related to how the algorithm works, in the sense that a different causal structure will result into a different outcome for the model. Without some assumptions on the nature of the DGP, however, the proposed statistical test will not uncover the true causal structure, and the results will be inconclusive. The assumptions that you mention are very relevant to interpret the results of the algorithm, even if they do not explecitly appear in the pseudo-code.
> >
> > >In the first equation on page 4, what is $N$?
> >
> > Thank you for spotting this typo! In fact, consistent with the notation used in the corresponding section of our paper, the correct equation for NAVAR is $$Y_T = \beta + \sum_{i=1}^{m} f_{i} (X^{i}_{T - \kappa:T-1}) + \eta_T, $$ where $m$ is number of nodes.
> >
> > >In equation 3, what is $n$? Should it be $k$?
> >
> > To the best of our understanding, Eq. 3 contains neither $n$ nor $k$. Can you please elaborate on this question, so that we can calrify your doubt?
> >
> > >I felt lost that in equation 3, $g^0_0$ and $g^i_0$ equal to the same conditional expectation as $\alpha^0_0$ and $\alpha_0^i$ in the second point in section 5.1. Are they the same things?
> >
> > Yes that's correct! This is just a nice coincidence of the application of the doubly robust score $\varphi( \theta, \boldsymbol \eta) := m(\boldsymbol{V}; g ) + \alpha(\boldsymbol{X})\cdot(Y - g(\boldsymbol{X})) - \theta$ (Equation 2 in the paper) to our particular setting, where the linear moment functional takes the form $m(\boldsymbol{V}; g )=Yg$ (proof of theorem 4.2 in the Appendix). Consequently, the Riesz Representer of this specific functional is $\alpha(\boldsymbol{X})=\mathbb{E}[Y\mid \boldsymbol{X}]$. Actually, in our experiments we have used explicitely this fact and since the Riesz representer $\alpha(\boldsymbol{X})$ coincides with the regressor $g(\boldsymbol{X})$ we can train only one model (we have ablated on this matter, i.e training two separate models for $\alpha(\boldsymbol{X})$ and $g(\boldsymbol{X})$, or just use a single model for both, and experimentally found out that there was no difference).
> >
> > However, in general the Riesz Representer $\alpha(\boldsymbol{X})$ does not coincide with the regressor $g(\boldsymbol{X})$; it does not even have an elegant explicit closed form. Moreover, the way of calculating it can be very different; for example, in [1],it is the minimizer of a particular loss, the Riesz Loss: $\widehat{\alpha} = argmin_{\alpha\in {\cal A}_n} \mathbb{E}_n[\alpha(Z)^2 - 2m(W; \alpha)]$.
> >
> > Since in practice $\alpha(\boldsymbol{X})$ and $g(\boldsymbol{X})$ are separate quantities, and in order to avoid confusion, we use different notation.
> >
> > >As the appendix states, $k$ ranges from 3 to 7. What is the value of $k$ used in each experiment? Is the algorithm output sensitive to the value of $k$?
> >
> > In all of the experiments reported in the paper, we have set k=5, with the only exception of figure 3 in the Appendix (page 29), where we used k=10. In this last experiment, where we test the performance of our algorithm in the low data regime, there are some settings where only as few as 5 (out of 46 in total) trajectories are used, so the number of observations is quite limited. In that case, we experimentally observed it is better to use k=10 folds (i.e each model training uses 90% of the data) than k=5 folds (i.e each model training uses 80% of the data). In all other settings, we experimentally observed that using k=10 yields the same performance compared to k=5. Moreover, we also found k=5 outperforms k=3.
> >
> > But even in the cases where some configuration for k is better compared to another one, the difference in performance (of the AUROC metric) is usually less than 5%. So the output is not very sensitive to the value of k.

---

> > > ### Author Response · Authors · 2023-11-17
> > > **Clarification - Continued II**
> > >
> > > >The term "trajectories" means the time series, correct? In Fig.2, what does $N_{feat}$ represent? Is $N_{feat}$ indicative of the number of trajectories? Additionally, in Fig.3, all the algorithms exhibit improved performance with an increase in the number of trajectories. Could you provide a brief explanation for this trend? Moreover, why does Fig.3 depict the performance in low-sample regimes, and how are "low-sample regimes" reflected in the Fig.3?
> > >
> > > We use the term "_trajectories_" only when we refer to the Dream3 dataset. In that context, _trajectory_ means group (i.e set) of time series. More specifically, the Dream3 challenge consists of 5 different tasks, where for each task we are given 46 groups of time series, and each group consists of 100 time series, each time series having length 21 timesteps. In this dataset, for a specific task, e.g E. Coli 1, we see how the measurements of 100 genes varie in time, after a perturbation to the system has occured at timestep t=0. In total, 46 different perturbations are applied to this system of 100 genes. Importantly, the causal dynamics of the gene system is the same in all these 46 different trajectories.
> > >
> > > In Fig.2, $N_{feat}$ actually is the same as $m$, the number of potential causes.
> > >
> > > Regarding Fig.3, based on the above explanation of trajectories, it is clear that fewer trajectories implies smaller number of observations used for causal inference. For example, when we have  5 trajectories, this means that we have observed the system of 100 genes in 5*21=105 timesteps in total, while 46 trajectories means that we have observed the system of 100 genes in 46*21=966 timesteps in total.
> > >
> > > >Is there a specific reason for using only one baseline algorithm in the experiments presented in the appendix?
> > >
> > > We only compare agains RHINO as this baseline is the main competitor and the best-performing algorithm. Other baselines are expected to perform worse, as already shown in previous related work and the results in Dream 3 benchmark (section 6.1).
> > >
> > > [1] Victor Chernozhukov, Whitney Newey, Victor Quintas-Martinez, and Vasilis Syrgkanis. RieszNet and ForestRiesz: Automatic debiased machine learning with neural nets and random forests. In Proc. of ICML, pp. 3901–3914, 2022.
> > >
> > > [2] Gong, Wenbo, Joel Jennings, Cheng Zhang, and Nick Pawlowski. “Rhino: Deep Causal Temporal Relationship Learning with History-dependent Noise.” In The Eleventh International Conference on Learning Representations. 2022.
> > >
> > > [3] Alex Tank, Ian Covert, Nicholas Foti, Ali Shojaie, and Emily Fox. Neural granger causality for nonlinear time series. stat, 1050:16, 2018.
> > >
> > > [4] Roxana Pamfil, Nisara Sriwattanaworachai, Shaan Desai, Philip Pilgerstorfer, Konstantinos Geor- gatzis, Paul Beaumont, and Bryon Aragam. Dynotears: Structure learning from time-series data. In International Conference on Artificial Intelligence and Statistics, pp. 1595–1605. PMLR, 2020.
> > >
> > > [5] Saurabh Khanna and Vincent YF Tan. Economy statistical recurrent units for inferring nonlinear granger causality. arXiv preprint arXiv:1911.09879, 2019.

---

> > > > ### Author Response · Authors · 2023-11-20
> > > > **Anymore Concerns?**
> > > >
> > > > Dear Reviewer,
> > > >
> > > > We sincerely appreciate the time you dedicated to evaluating our work. We would like to inquire if our rebuttal adequately addressed your concerns and if we can provide any additional information that you think is necessary.
> > > >
> > > > Thank you for your consideration!

---

> ### Comment · Reviewer_95xj · 2023-11-22
>
> "In equation 3, what is n? Should it be k?" Sorry that I made a mistake, and it should be equation 4.
>
> Thank you for the detailed clarification; my questions have been well answered.

---

> > ### Author Response · Authors · 2023-11-22
> > **Further Clarification**
> >
> > We thanks the reviewer for engaging in the discussion.
> >
> > > "In equation 3, what is n? Should it be k?" Sorry that I made a mistake, and it should be equation 4.
> >
> > $n$ is the number of samples. The equation 4 is about the strong consistency statement. Another way to represent that is by  $(\theta^0 - \theta^i) \overset{\mathrm{d}}{\rightarrow} \mathcal{N}(0, \frac{\sigma^2}{n})$ (where $\overset{\mathrm{d}}{\rightarrow}$ denotes convergence in distribution), indicating that the variance of the estimation error tends to zero as the sample size ($n$) approaches infinity.
> >
> > We are openly receptive to further engaging discussions and appreciate that if the reviewer could increase the score, now that we have addressed all concerns carefully.

---

> > > ### Comment · Reviewer_95xj · 2023-11-22
> > >
> > > Q3 can be described in a more precise manner as follows. $\theta^0-\theta^i$ does not incorporate $n$. How might one obtain the distribution of $\theta^0-\theta^i$ concerning convergence as $n$ approaches infinity, considering the absence of $n$ within the expression?

---

> ### Author Response · Authors · 2023-11-23
> **Reply to reviewer**
>
> Thank you for pointing this out. Indeed, $\theta^0$ and $\theta^i$ are sample estimates of the true parameters $\theta^0_0 = \mathbb{E}[Y_T \cdot g^0_0 ]$ and $\theta^i_0 = \mathbb{E}[Y_T \cdot g^i_0 ]$ respectively and  of course their distribution depends on $n$. The only reason we did not make this dependence explicit in the notation was simplicity, in order not to overload variable symbols with too many subscripts and superscripts.

---

### Official Review · Reviewer_uCS8 · 2023-11-01

**Soundness:** 3 good
**Presentation:** 2 fair
**Contribution:** 3 good
**Rating:** 6
**Confidence:** 3

**Summary:**

This paper proposes an algorithm for doubly robust structure identification for Granger causality. It also provides asymptotical guarantees that the proposed method can discover the direct causes even when there are cycles or hidden confounding and that the algorithm has $\sqrt(n)$-consistency.

**Strengths:**

The proposed doubly robust structure identification for Granger causality is novel, as far as I know. The paper also provides identifiability guarantees in the presence of cycles or hidden confoundings.

**Weaknesses:**

The paper did not analyze or give an intuition why the proposed method allows the existence of cycles or hidden confoundings.

**Questions:**

Why does the proposed method allow the existence of cycles or hidden confounding?

---

> ### Author Response · Authors · 2023-11-17
> **Clarification**
>
> Thank you very much for taking the time to review our work and recognize its merits!
>
> > The paper did not analyze or give an intuition why the proposed method allows the existence of cycles or hidden confoundings. Why does the proposed method allow the existence of cycles or hidden confounding?
>
> To give an intuition of why we can handle hidden confounding and cycles, denote with $Y$ the target time series and with $\\{ X^1, \dots, X^n \\}$ the observed time series that are potential causes for $Y$. Using Axioms $A$-$C$, we prove that in this case, performing an intervention on all the variables  $\\{ X^1, \dots, X^n \\}$ is equivalent to conditioning (this fact is specific to our causal model and it is not true in general). This type of conditioning "cancels out" the effect of cycles or hidden confounding on $\\{ X^1, \dots, X^n \\}$.

---

> > ### Author Response · Authors · 2023-11-20
> > **Anymore Concerns?**
> >
> > Dear Reviewer,
> >
> > We sincerely appreciate the time you dedicated to evaluating our work. We would like to inquire if our rebuttal adequately addressed your concerns and if we can provide any additional information that you think is necessary.
> >
> > Thank you for your consideration!

---

### Official Review · Reviewer_WUkY · 2023-11-01

**Soundness:** 3 good
**Presentation:** 3 good
**Contribution:** 3 good
**Rating:** 6
**Confidence:** 3

**Summary:**

This paper presents a novel and efficient Doubly Robust Structure Identification from Temporal Data (SITD) algorithm, offering theoretical guarantees including $\sqrt{n}$-consistency. It establishes a technical connection between Granger causality and Pearl's time series framework, outlining the conditions under which the approach is suitable for feature selection and full causal discovery. The paper's theoretical insights highlight the algorithm's ability to handle non-linear cyclic structures and hidden confounders, even without relying on faithfulness or causal sufficiency. In extensive experiments, the approach demonstrates remarkable robustness, speed, and performance compared to state-of-the-art methods, making it a valuable contribution to causal discovery in various applications.

**Strengths:**

- They've introduced a doubly robust structure identification method for analyzing temporal data. It doesn't rely on strict faithfulness and causal sufficiency assumptions, making it versatile enough to handle general non-linear cyclic structures and hidden confounders.

- The innovative application of the double machine learning framework to Granger causality is a significant contribution.

- The paper is well-structured, maintaining a coherent and easily-followed flow from beginning to end.

- The paper extensively references related work, offering a comprehensive overview of prior research that not only provides valuable context for the study but also underscores the authors' profound understanding of the field.

**Weaknesses:**

- Regarding the "stationary causal relation" assumption, you mentioned that the results could potentially apply to models that do not meet this axiom. Have you formally demonstrated this claim in any specific section, or are you implying that the proof of Theorem 4.1 does not rely on this assumption?

**Questions:**

- How do you identify cyclic structures? Does Algorithm 1 have the capability to detect cyclic structures, and does this imply the presence of confounders?

- In your method, is the time lag $k$ fixed, or does it remain stationary but vary among different variables?

---

> ### Author Response · Authors · 2023-11-17
> **Clarification**
>
> Thank you for enumerating the strengths of our work. We address each of your concerns in the following and look forward to further discussion if you have additional feedback.
>
> > Regarding the "stationary causal relation" assumption, you mentioned that the results could potentially apply to models that do not meet this axiom. Have you formally demonstrated this claim in any specific section, or are you implying that the proof of Theorem 4.1 does not rely on this assumption?
>
> Under the stationarity assumption, we can test direct causal effects, by focusing on a target $Y_T$ for a fixed time T. If stationarity does not hold, then we will have to perform the test on $Y_t$, for all time steps t. This fact follows directly from our proof.
>
> > How do you identify cyclic structures? Does Algorithm 1 have the capability to detect cyclic structures, and does this imply the presence of confounders?
>
> Our algoirthm only identifies direct causal effects on a target variable of interest. It does not identify cycilc structure. However, as detailed in Axioms (A)-(B), our algorithm works even under confounding or cycles among the covariates. In other words, even if hidden confounders and cycles are present, we are still able to uncover the direct causes.
>
> > In your method, is the time lag $k$ fixed, or does it remain stationary but vary among different variables?
>
> In general, we only assume stationarity, and $k$ may vary among different variables. However, in our experimental framework we set $k$ to always be the same for simplicity. Specifically, in all our experiments, time lag is equal to 2. This is quite a standard value accross the literature [1,2,3,4].
>
>
> [1] Gong, Wenbo, Joel Jennings, Cheng Zhang, and Nick Pawlowski. “Rhino: Deep Causal Temporal Relationship Learning with History-dependent Noise.” In The Eleventh International Conference on Learning Representations. 2022.
>
> [2] Saurabh Khanna and Vincent YF Tan. Economy statistical recurrent units for inferring nonlinear granger causality. arXiv preprint arXiv:1911.09879, 2019.
>
> [3] Roxana Pamfil, Nisara Sriwattanaworachai, Shaan Desai, Philip Pilgerstorfer, Konstantinos Geor- gatzis, Paul Beaumont, and Bryon Aragam. Dynotears: Structure learning from time-series data. In International Conference on Artificial Intelligence and Statistics, pp. 1595–1605. PMLR, 2020.
>
> [4] Alex Tank, Ian Covert, Nicholas Foti, Ali Shojaie, and Emily Fox. Neural granger causality for nonlinear time series. stat, 1050:16, 2018.

---

> > ### Author Response · Authors · 2023-11-20
> > **Anymore Concerns?**
> >
> > Dear Reviewer,
> >
> > We sincerely appreciate the time you dedicated to evaluating our work. We would like to inquire if our rebuttal adequately addressed your concerns and if we can provide any additional information that you think is necessary.
> >
> > Thank you for your consideration!

---

### Author Response · Authors · 2023-11-17
**Author Rebuttal**

We express our gratitude to the reviewers for their interest and insightful questions as well as their perceptive and constructive remarks. We have tried our best to address their concerns in our responses. We look forward to receiving additional feedback and eagerly anticipate any further discussions to enhance the clarity and scope of our work.

In this manuscript, we introduced a computationally fast and easy-to-implement doubly robust causal parents identification from temporal data algorithm (SITD) with theoretical guarantees enjoying fast $\mathcal{O}(n^{-1/2})$ parametric rate. Our algorithm doesn't need faithfulness or causal sufficiency while allowing for general non-linear cyclic structures and also the presence of hidden confounders, which is a major improvement in the literature.

We believe that the reviewers may have overlooked the computational simplicity of our method when comparing it to existing literature. We kindly request them to consider the computational aspects in their evaluations. While existing baselines such as PCMCI+ require exponential time and deep learning-based approaches like Rhino demand substantial computational resources, specifically GPUs, our method, employing only $\mathcal{O}(m)$ regressions, yields competetive (better in most cases) results in less than a minute, even for massive causal graphs. For further details, please refer to Appendix A.9 and A.10.3.

We postulate that our work is a significant milestone in provably learning causal structures of time series under minimal assumptions (that still preserve identifiability of the causal structure from solely observational data). We hope this work estabilishs a firm starting point for future directions.

Our postulations are supported by the reviewers. Reviewer biK3 asserted that "_The approach is __very novel__ to the best of my knowledge_". Reviewer WUkY pointed out that "_They've introduced a doubly robust structure identification method for analyzing temporal data. It doesn't rely on strict faithfulness and causal sufficiency assumptions, making it __versatile enough__ to handle general non-linear cyclic structures and hidden confounders_" or elsewhere stated  that "_The __innovative__ application of the double machine learning framework to Granger causality is a __significant contribution___". Reviewer 95xj maintained "_The authors offer a discussion connecting Granger's causality with Pearl's framework, which is __thought-provoking___". Reviewer fyvt mentioned "The research question addressed in the paper is of __very high importance.__". Finally, reviewer uCS8 mentioned "_The proposed doubly robust structure identification for Granger causality is __novel__, as far as I know._".

Lastly, it is worth noting that we are openly receptive to further engaging discussions.

---

### Meta-Review · Area_Chair_hNEZ · 2023-12-06

**Metareview:**

The paper proposes a doubly robust method for Structure Identification from Temporal Data (SITD), providing theoretical guarantees for asymptotically recovering the true causal structure, even in scenarios with cyclic data and potential confounding.

pros:
+ The paper works on an important problem of structure identification in time series data with potentially cyclic structure.
+ The algorithm is versatile, handling general non-linear cyclic structures and hidden confounders without relying on strict faithfulness or causal sufficiency assumptions.

cons:
+ lack of a real-world application, potentially limiting its practical relevance
+ the paper's claim about the algorithm's use for full causal discovery lacks detailed discussion
+ some concerns around theoretical soundness of the proposal

**Justification For Why Not Higher Score:**

concerns around both theoretical soundness and empirical studies

**Justification For Why Not Lower Score:**

NA

---

### Decision · Program_Chairs · 2024-01-16

Reject